# Back to the Future: Agricultural Booms, Busts, and Diversification in Maine, USA, 1840–2017

Aaron Kinyu Hoshide [1,2]

1   College of Natural Sciences, Forestry and Agriculture, The University of Maine, Orono, ME 04469, USA;
    aaron.hoshide@maine.edu; Tel.: +1-207-659-4808
2   AgriSciences, Universidade Federal de Mato Grosso, Sinop 78555-267, MT, Brazil

**Abstract:** In temperate forested regions, historical agricultural production and value have been characterized by booms and busts. Agricultural diversification can encourage more stable agricultural development in the future. Agricultural Census and Survey data from 1840 to 2017 were used to estimate crop and livestock species' product production and value for Maine, USA. These data were also used to calculate agricultural diversity indicators over time such as species richness, relative abundance, effective number of species, species diversification index, evenness, Shannon-Weiner index, and composite entropy index. Maine's historical grass-based livestock systems included crops raised to feed livestock from the state's establishment until the 1950's. Since the 1950's, production and value of livestock commodity products (e.g., meat chicken, eggs) have busted after initial booms. Three categories where diversity indicators have become more favorable since the 1950's in Maine include livestock, livestock forage/feed, and potatoes and potato rotation crops. Mixed vegetables, fruits, nuts, and specialty crops as a category have had diversity increases during the 1970's back-to-the-land movement and over the past two decades. Floriculture, propagation, and X-Mas trees as a category have witnessed volatile diversity indicator changes over time. Past diversification strategies can inspire farmers to go "back to the future" to improve sustainability.

**Keywords:** agricultural development; sustainability; diversity indexes; cultivars; livestock breeds; Maine

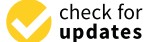



## 1. Introduction

Forests contribute to global biodiversity of terrestrial species [1] especially when disturbances to forest ecosystems are moderate [2] and trees and understory plants are more diverse [3]. Diversity for temperate forests is typically greater immediately following clearcutting and during a forest's terminal and decay stages after 200 to 300 years [4]. Maine USA forests are currently in intermediate successional stages due to industrial logging requiring more active management to increase biodiversity [5]. Historical logging in Maine (Figure 1) cleared enough land to allow for relatively high percentages of the southern and central (~70%), western (~40%), and northern (~20%) parts of the state to be used for agriculture from 1860 to 1920 [6] with a sharp decline in agricultural farmland between 1950 and 1970 (Figure 2). Compared to estimating the economic value of forest biodiversity which has focused on whole ecosystems or species within such ecosystems [7], relatively little research has been done on measuring crop/livestock diversity and economic value of agricultural systems within temperate forested areas over longer historical time periods. Crop diversity for commodity field crops in the USA has declined [8,9], peaking around 1960 [9], and has been positively influenced by irrigation [10]. More nuanced analyses are needed evaluating diversity and value of vegetables, fruits, nuts, specialty crops, and livestock as influenced by farming booms and busts as well as national/regional specialization and diversification.

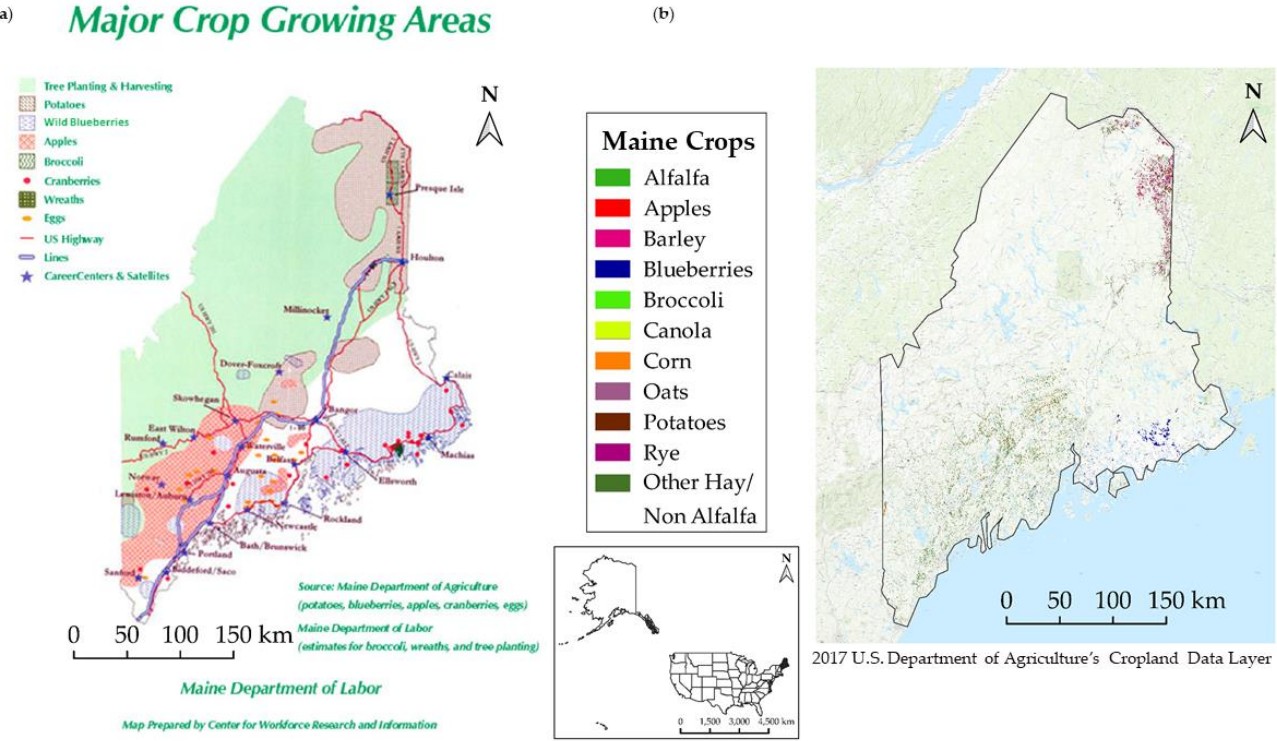

**Figure 1.** (**a**) Major crop growing areas and (**b**) major crops in Maine, USA. Reprinted/adapted with permission from Refs. [11,12]. 2022, Maine Department of Agriculture and Maine Department of Labor [11] and U.S. Department of Agriculture, National Agricultural Statistics Service [12].

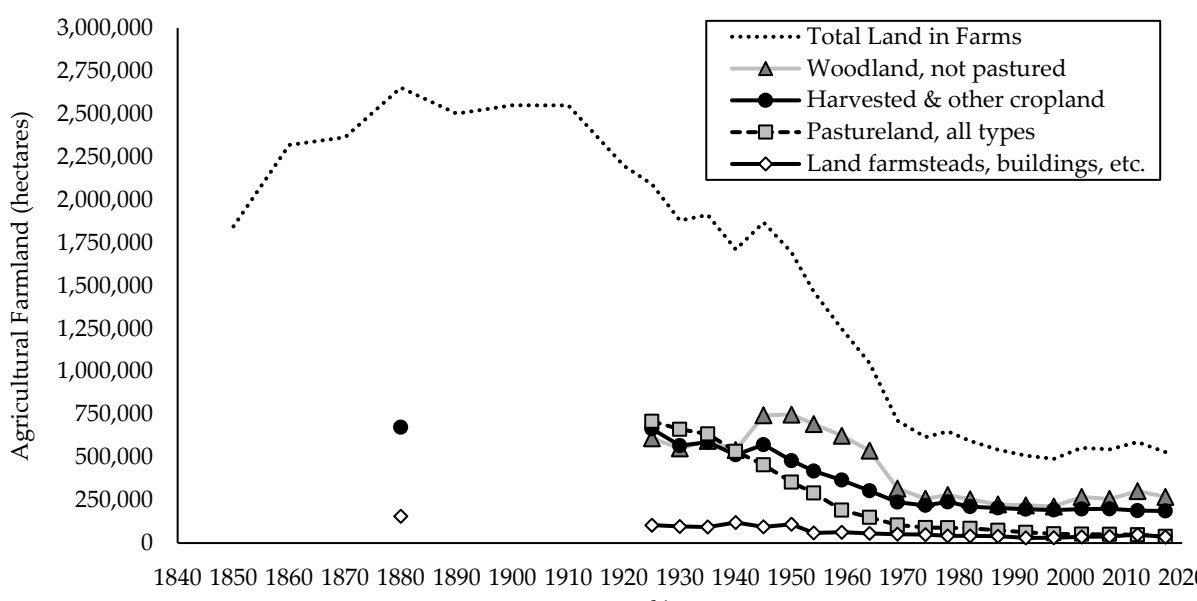

**Figure 2.** Agricultural farmland area (hectares) from 1840 to 2017 for Maine, USA.

Agricultural booms and busts over the past century in the USA have been driven by global trade and macroeconomics. The USA has had short-term and medium-term agricultural booms/busts from 1900 to 2015 driven by international export markets from (a) 1910–1930, (b) 1970–1990, and (c) 2000–2010 [13,14]. Economic downturns associated with these boom and bust cycles were attributed to banks aggressively lending after opening during the boom with subsequent collapse in farmland value during the bust (1910–1930) [15], the 1973–1974 oil embargo and stagflation (1970's), and the 2007–2008

Great Recession where cereals and vegetable oil were impacted but livestock was not [16]. The Midwest Corn Belt 1970–1990 boom and bust during the 1980's was triggered by a drop in export demand combined with increasing interest rates [17]. In Saskatchewan, Canada during the 1970–1990 boom and bust, the bust was delayed by crop and livestock diversification combined with expectations of temporary rather than extended down cycles. However, credit, human capital, and technical knowledge were required to diversify more into beef, pulses, and oilseeds [18].

Agricultural diversity has peaked and declined in regions and countries around the world throughout the 20th century. For example, Simpson's Index of diversity peaked in the 1950's and declined to 1992 in West Punjab, India [19]. The agricultural industry in the USA from after World War II until the mid-1970's has gotten more specialized [20] where diversity (D) measured as effective crop species weighted by the Shannon-Weiner Index has declined in the USA since the 1960's [9]. Farm-level specialization/diversification revolve around farm economics. Agricultural specialization is driven by economies of scale which maximizes production of one commodity for a specific degree of capital investment. Specialization of farms and entire agricultural industries are susceptible to agglomeration in areas of the world that provide comparative advantages of production, processing, and marketing. Areas that become less competitive lose out. Farms in dying industries must either sustainably intensify or diversify to remain viable [21]. Within-farm diversification can also be triggered by unfavorable conditions [22], such as lower market prices of agricultural products produced as well as higher inputs costs [23]. Farm characteristics that support the ability to diversify include having enough labor slack [22] and spouse/family labor [23].

The goal of this research is to estimate the production and value of commodity field crops, vegetables, fruits and nuts, specialty crops, and livestock in Maine USA from 1840 to 2017. USDA Agricultural Census and Survey data [24] was analyzed over this time frame in order to delineate Maine's boom and bust cycles of farming which have resulted in efforts to diversify its food systems. Thus the specific objectives of this study are to (1) determine production and inflation-adjusted value peaks for all agricultural crops and livestock over this 177 year period, (2) calculate the diversity of these agricultural enterprises using common ecological diversity indicators, and (3) explain recent diversification trends in Maine as responses to boom and bust of key agricultural commodities. Past and current agricultural diversification in Maine can serve as models on how to go "Back to the Future" to better diversify agricultural systems in other temperate regions.

## 2. Materials and Methods

### 2.1. Determining Historical Agricultural Production and Value

In order to identify boom and bust periods for both crops and livestock species in Maine, the production and value for each agricultural product produced from these species had to be calculated using historical data. Maine USA livestock numbers and crop area, livestock/crop farm numbers, agricultural product yields, and values were downloaded and analyzed for 29 Census of Agriculture years starting in 1840 and ending in 2017 [24]. Crops and livestock production required English to metric conversions for weights of farm-gate products produced in any given Census year. Since livestock forages and feeds had different dry matter (DM) percentages, total forage and feed production was calculated on a dry matter basis using previous assumptions for dry hay, corn silage, alfalfa hay/silage [25], sorghum silage [26], and pumpkins used for feed [27]. Root crops for livestock feed were assumed to have the same DM as forage turnips [28].

Livestock also required estimating animal live weights, carcass weights (if slaughtered), and weights of products produced (e.g., milk, meat, fiber) using appropriate conversion factors. If USDA data [24] did not provide animal product production but rather only animals sold, it was assumed this was for meat and not for breeding. Annual assumed meat production (e.g., beef, pork, poultry, rabbits, etc.) was estimated as:

$$\text{Meat production} = \text{Animals sold} \times (\text{Live weight/animal} \times \text{Dressing percentage}) = \text{Animals sold} \times \text{Carcass weight/animal} \tag{1}$$

Cattle live weights and carcass weight conversions were based on past research in Maine [25,29]. However, beef products could not be estimated since animals sold were not distinguished between feeder, slaughtered, and live breeding cattle in USDA statistics [24]. Pork livestock weight and dressing percentage were based on past work with local producers (Aaron K. Hoshide, unpublished data). Sheep and goat dressing percentage and/or live weights were from [30–33]. Horse live weight was based on [34]. Similar assumptions for poultry live weight and/or dressing percentage were used for broiler chicken [35–37], turkey [35,37], goose, duck [35], pheasant [38], guinea fowl [35], quail [39,40], pigeon [41], and emu [42]. Similar assumptions were used for bison [43], tame deer [44], and rabbit [45–47]. Conversions were used for chukar [48], partridge [49], ostrich [50], rhea [51], and chinchilla [52,53], but these animals did not have enough data to delineate boom/bust periods of production.

Non-meat animal products included milk from cows, chicken eggs, wool from sheep, and mohair from goats. Earlier Census of Agriculture years required volume to weight conversion for milk and other dairy products [54]. Live weights of laying chickens were from [55]. Egg production was available for Census years from 1880 to 1964 [24], but had to be estimated for 1969 to 2017 by multiplying the number of layers [24] by the average annual egg production per layer in Maine (1969–2007), the average in the nearby states of Massachusetts and Vermont (2012), and Vermont (2017) [56]. Sheep fleece and mohair goat fiber weights per animal were used [26].

Crop and livestock values were either available [24] or were estimated based on total production of crops/livestock multiplied by products' per unit prices. All nominal prices and values in any Agricultural Census year between 1840 and 2012 were converted to real prices and values in 2017 USD. Such adjustments for inflation were to a base year of 2017 using USA commodity specific Producer Price Indexes (PPI) when possible. Missing commodity specific PPI data from 1926 to present used composite PPI for four categories: (1) fruits and melons, fresh/dry vegetables and nuts, (2) grains, (3) hay, hayseeds, and oilseeds, and (4) slaughter livestock. Missing commodity specific PPI data from 1913–1925 used the farm products composite PPI. Crop and livestock categories without specific PPI used the all commodities PPI for 1840–2017 [57]. If the farm-gate price for a product was not available for Maine in a particular year, then a regional (e.g., New York State) or USA national price was used from USDA Agricultural Survey data [24]. The value of unthreshed oats harvested to feed livestock (1925–1950) was estimated as the sum of both grain and straw values. Oat straw prices were obtained from Andrew Plant, University of Maine Cooperative Extension in 2017.

### 2.2. Calculating Agricultural Diversity Indicators

USDA Agricultural Census data for crop and livestock species numbers [24] were used to determine the diversity of major categories of crops and for livestock. Seven diversity indicators were calculated from these data for four crop categories: (1) mixed vegetables, fruits, nuts, and specialty crops, (2) potatoes and annual crops rotated with potatoes, (3) floriculture, propagation, seeds, and Christmas (X-Mas) trees, and (4) livestock forage/feed crops. These seven diversity indicators were also calculated for a fifth category for all livestock species. Three of these seven indicators were related to species number (richness, effective number of species) and relative abundance. The remaining four indicators were measured on a scale of 0 (no diversity) to 1 (highest diversity) and included the Species Diversification Index, evenness, the Shannon-Weiner Index, and the Composite Entropy Index.

### 2.2.1. Richness, Relative Abundance, and Effective Number of Species

Agricultural diversity can be measured by the number of crop/livestock species in a particular area. For ecological systems (e.g., forests, agro-ecological agriculture), alpha diversity measures within-species diversity, beta diversity contrasts diversity between different species, and gamma diversity measures the biodiversity across an entire area, region, or biome [2,4]. For crops and livestock, richness is the number of total crop cultivars or total livestock breeds in a given time period (e.g., Agricultural Census year) for a particular area (e.g., Maine, USA).

Relative abundance for a particular crop or livestock category was calculated as the percent by area/weight relative to all other categories. So for example for crops, the relative abundance for the category livestock forage/feed is the percent of total crop area this category makes up in a particular year relative to all other crop categories. The relative abundance of eggs is its percent of total product weight relative to other livestock product categories [58].

The effective number of species (e.g., crop, livestock) or *ENS* is richness (*R*) multiplied by the natural exponent of the negative Shannon-Weiner Index (*SWI*) of diversity:

$$ENS = R \times e^{-SWI} = R \times e^{-\sum_{i=1}^{s}(p_i \times log p_i)} \tag{2}$$

where $p_i$ is the proportion of species $i$ within the total number (s) of species within the crop or livestock category [8–10]. So *ENS* adjusts *R* by the both the number of species and relative proportions of species within an agricultural category such as crops or livestock. For example, if there are 10 crop (livestock) species with each species making up 10% of the total category area (live weight), then $e^{-SWI}$ equals 1, which means *ENS* = *R* × $e^{-SWI}$ = *R* × 1 = *R* in this particular case. However, if the number of species are <10 and/or if certain species make up a disproportionate percentage of the total category then $e^{-SWI}$ will be less than 1 and thus *ENS* < *R*. If species are more evenly distributed and/or if *R* > 10, then $e^{-SWI}$ can be greater than 1 and thus *ENS* > *R*.

### 2.2.2. Diversity Indexes

The four diversity indexes (0 to 1) evaluated were Species Diversification Index for both crop and livestock species, evenness, Shannon-Weiner Index, and the Composite Entropy Index. Species Diversification Index (*SDI*) equals one minus the Simpson's Index (*SI*) in agro-ecology (0 to 1) or one minus the sum of squared proportions of crop/livestock species:

$$SDI = 1 - SI = 1 - \sum_{i=1}^{s} p_i \tag{3}$$

where $p_i$ is the proportion of species $i$ within the total number (s) of species within the crop or livestock category [59]. *SI* is identical to the economic Herfindahl Index (HI). HI measures the degree of market concentration for businesses within a particular industry as the sum of squared market shares. When using market share proportions (versus percentages), HI ranges from 0 to 1 (1 to 10,000 when using percentages). For a monopoly dominating an entire industry (1 = 100%), HI = $1^2$ = 1 × 1 = 1. For a perfectly competitive industry with a large number of equally sized firms, HI approaches 0.

Evenness (*E*) is how balanced crop or livestock species are in a particular category. So *E* is lower if fewer species make up a disproportionally large percent of the total category [58]. *E* is calculated as *SI* divided by the natural log of the richness of species and ranges from 0 to 1:

$$E = \frac{SI}{ln R} \tag{4}$$

As specified as part of Equation (2), the Shannon-Weiner Index (*SWI*) is the negative value of the sum of squared proportions times the log of proportions:

$$SWI = - \sum_{i=1}^{s} (p_i \times log p_i) \tag{5}$$

with values ranging from 0 to positive 1. The Composite Entropy Index (*CEI*) weights the *SWI* by 1 − (1/*N*) where *N* equals the total number of crops or livestock species in particular agricultural category. *CEI* is defined as:

$$CEI = -[\sum_{i=1}^{s}(p_i \times log\, p_i)] \times [1 - \frac{1}{N}] \quad (6)$$

where *N* equals the total number of species of crops or livestock. So if there is only one species, then 1 − (1/*N*) = 1 − (1/1) = 1 − 1 = 0 so *CEI* will equal 0 (no diversity). If there is a very large number of species, then *CEI* will be much closer in value to *SWI* [60,61].

## 3. Results

### 3.1. Agricultural Booms and Busts

Major crop production categories of potatoes, grains and oilseeds, and dry matter of livestock forages/feeds went through different boom and bust periods with production peaking in different years. Potato production in Maine USA peaked at 2,176,798 metric tons (t) in 1950 (Figure 3a) on 58,028 hectares (ha) (Figure 3b). For grains and oilseeds, production peaked in 1860 at 140,059 t harvested from 91,118 ha with a more recent rebound since 1970 to 95,885 t grown on 19,278 ha in 2017 (Figure 4a). Grain/oilseed production in 2017 was only 68.5% of production and 21.2% of crop area compared to the historical production peak in 1860. Agricultural Census data for Maine [24] did not have comprehensive production and value data for mixed vegetables/fruits (Table 1) except for the year 1900. Historical trends in other crops included declines in orchard fruit and dry beans, increase in berries (Figure 4b), and a brief boom/bust period for sugar beets around 1969 (Figure 5).

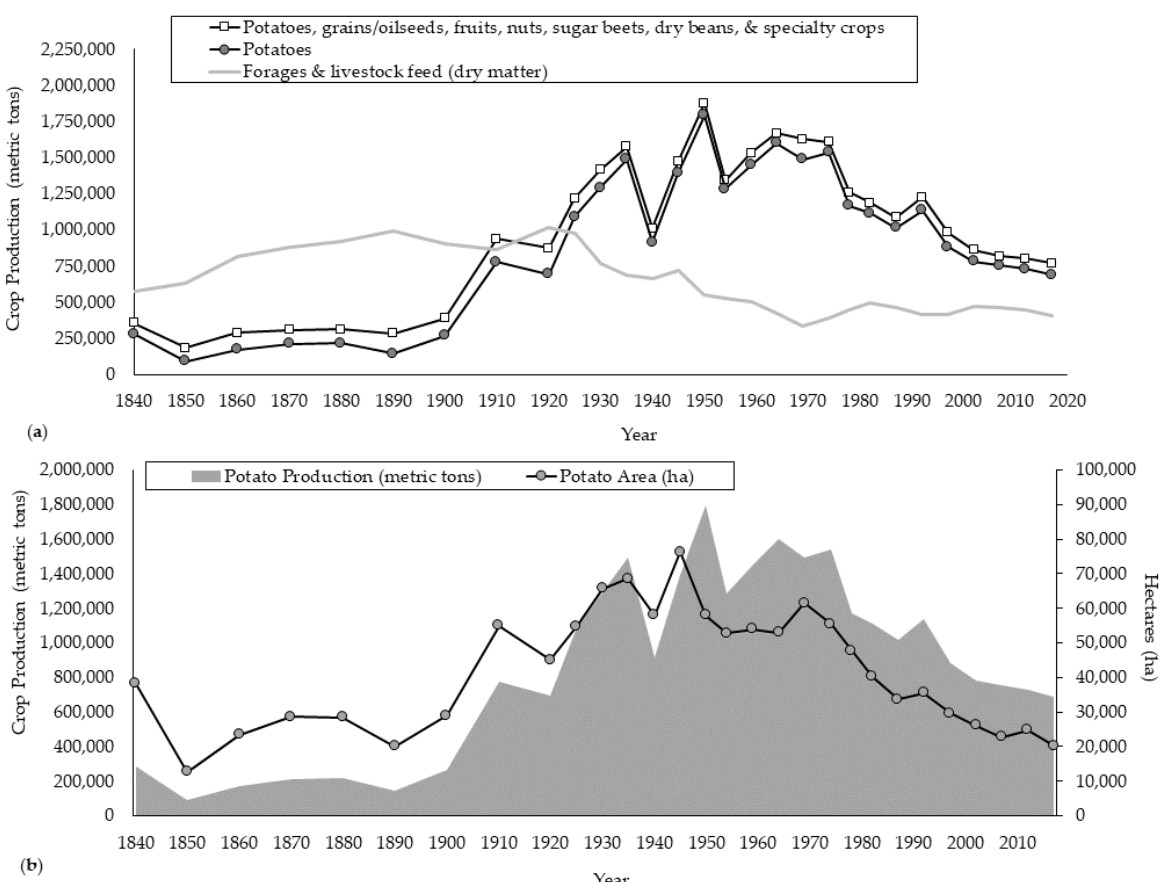

**Figure 3.** (**a**) Crop category weights in metric tons (t) and (**b**) potato production in metric tons (t) and area (hectares) from 1840 to 2017 for Maine, USA.

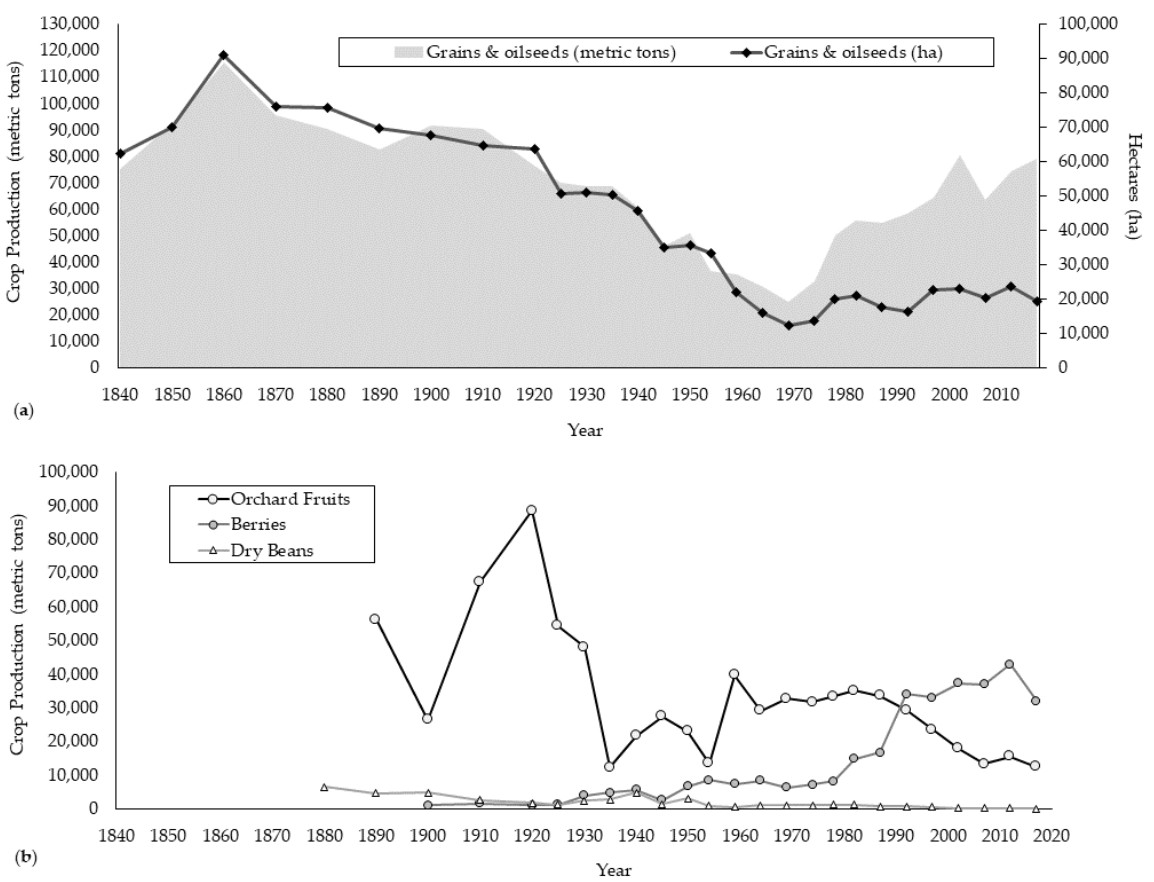

**Figure 4.** (**a**) Grain/oilseed production in metric tons (t) and area (hectares) and (**b**) orchard fruit, berry, and dry bean production in metric tons (t) from 1840 to 2017 for Maine, USA.

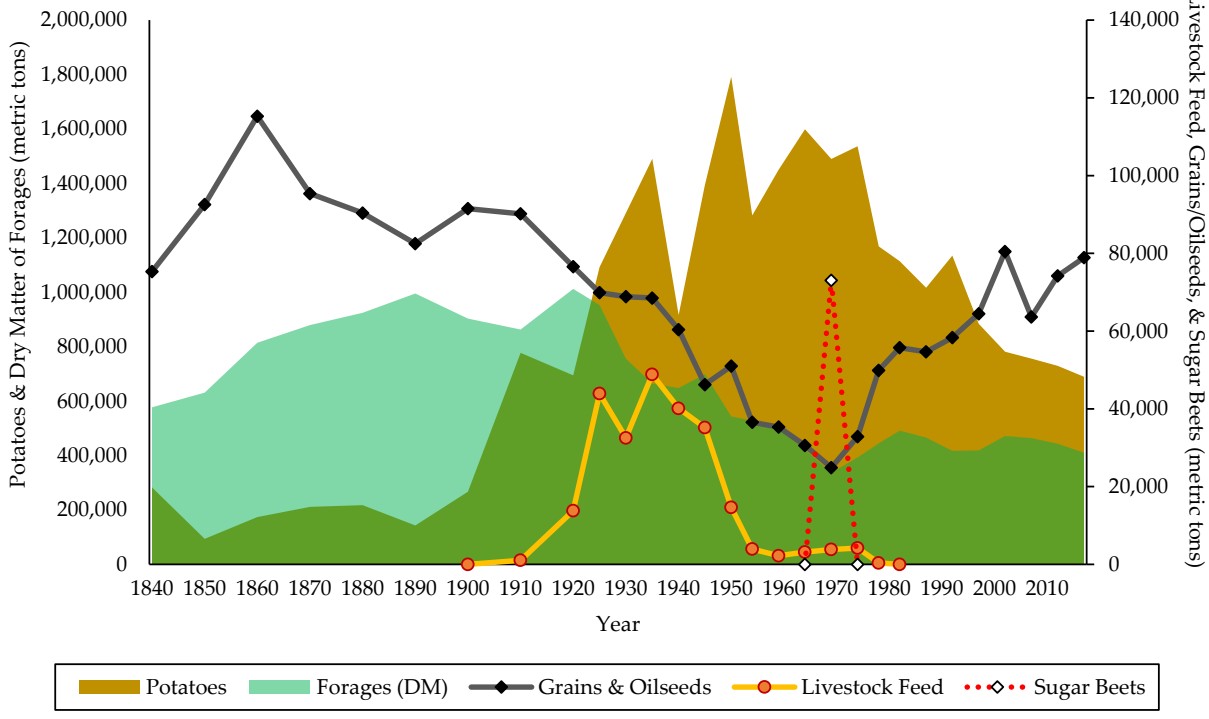

**Figure 5.** Potato, forage dry matter, livestock feed, and sugar beets in metric tons (t) from 1840 to 2017 for Maine, USA.

**Table 1.** Earlier agricultural booms and busts for livestock and crop products in Maine, USA, calculated or summarized from publicly available USDA-NASS statistics. Reprinted/adapted with permission from Ref. [24]. 2022, U.S. Department of Agriculture.

| Agricultural Category and Enterprise | Latin Name (*Genus Species Subspecies*) | Boom/Bust Years (Start–*Peak*–End) | - - - -Peak Production Year Estimate- - - - | | | | 2017 Percent of Peak Prod. Year | |
|---|---|---|---|---|---|---|---|---|
| | | | Farms | Area (ha) | Product [1] (metric t) | Real Value (2017 USD) | Product | Value |
| **LIVESTOCK** | | | | | | | | |
| Hogs | *Sus domesticus* | 1840–***1840***–1959 | 35,101 | - | 10,721 | 3,423,882 | 9.9% | 55.3% |
| Sheep (wool) | *Ovis aries* | 1840–***1880***–1900 | 36,396 | - | 1259 | 11,931,087 | 3.5% | 0.3% |
| Geese | *Anser* spp. *domesticus* | 1880–***1890***–1910 | 1911 | - | 13 | 71,179 | 3% | 1.7% |
| Horses [2] | *Equus ferus caballus* | 1840–***1890***–1945 | 47,420 | - | - | - | - | - |
| Pigeons | *Columba livia domestica* | 1910–***1910***–1910 | 287 | - | 0.40 | 6937 | 43.3% | 75.7% |
| Guinea Fowl | *Numida meleagris* | 1910–***1910***–1910 | 1073 | - | 3.64 | 27,828 | 10.6% | 2.3% |
| Angora Goat | *Capra hircus aegagrus* | 1900–***1910***–1920 | 39 [3] | - | 0.29 [3] | 3152 [3] | 85.4% [3] | 22.8% [3] |
| **FORAGES/ FEED** | | | | | | | | |
| Seed | *Fabaceae/Poaceae* family | 1850–***1860***–1890 | 9818 | 15,227 | 1460 | 6,223,061 | 1.1% | 1.8% |
| Forage hay | *Poa* spp. & others | 1840–***1920***–1959 | 46,790 | 496,582 | 1,094,358 | 212,351,008 | 30.2% | 22.8% |
| Root crops | Not specified | 1910–***1920***–1950 | 1872 | 633 | 8411 | 1,551,150 | 0% | 0% |
| Oats unthrsh. | *Avena sativa* | 1925–***1925***–1950 | 6272 | 6582 | 20,724 | 3,653,218 | 0% | 0% |
| Corn hogged | *Zea mays* | 1920–***1935***–1978 | 2273 | 1512 | 34,336 | 4,028,874 | 0% | 0% |
| Pumpkins | *Cucurbita pepo* | 1940–***1940***–1945 | 31 | 8 | 58 | 8850 | 0% | 0% |
| **GRAIN/ OILSEED** | | | | | | | | |
| Rye | *Secale cereale* | 1840–***1840***–1870 | 5335 | 5409 | 3504 | 1,171,145 | 59.1% | 32.2% |
| Wheat | *Triticum aestivum* | 1840–***1840***–1935 | 23,487 | 22,598 | 23,083 | 13,002,123 | 1.6% | 0.7% |
| Flaxseed | *Linum usitatissimum* | 1850–***1850***–1910 | 98 | 124 | 15 | 7724 | 0% | 0% |
| Corn | *Zea mays* | 1840–***1850***–1920 | 36,109 | 22,852 | 44,453 | 7,052,906 | 68.2% | 68.9% |
| Barley | *Hordeum vulgare* | 1840–***1860***–1900 | 14,369 | 13,059 | 17,464 | 4,674,916 | 144% | 76.7% |
| Buckwheat | *Fagopyrum esculentum* | 1840–***1900***–1954 | 9727 | 10,235 | 11,046 | 3,725,345 | 0.4% | 1.0% |
| Small Grains | Not specified | 1925–***1950***–1964 | 350 | 2062 | n/a | n/a | 0% | n/a |
| **VEGETABLES** | | | | | | | | |
| Dry peas | *Pisum sativum* | 1880–***1880***–1920 | 8531 | 1784 | 1495 | 1,736,056 | 0% | 0% |
| Dry beans | *Fabaceae* Family spp. | 1880–***1880***–1997 | 30,005 | 6852 | 4945 | 6,229,964 | 1.3% | 0.6% |
| Sweet potato | *Ipomoea batatas* | 1890–***1890***–1890 | 6 | 1.6 | 6 | 4309 | 142% | 126.3% |
| Sweet corn | *Zea mays* | 1900–***1930***–1950 | 7153 | 6654 | n/a | n/a | n/a | n/a |
| Cabbage | *Brassica oleracea capitata* | 1900–***1935***–1969 | 1273 | 242 | n/a | n/a | n/a | n/a |
| Squash | *Cucurbita* spp. | 1930–***1940***–2017 | 494 | 478 | n/a | n/a | n/a | n/a |
| Green beans | *Phaseolus vulgaris* | 1920–***1945***–1978 | 2397 | 1616 | n/a | n/a | n/a | n/a |
| **FRUITS** | | | | | | | | |
| Grapes | *Vitis vinifera* | 1900–***1900***–1950 | 4350 | 20 | 125 | 74,441 | 118% | 192.5% |
| Apricots | *Prunus armeniaca* | 1910–***1910***–1910 | 48 | 7 | 0.57 | 605 | 0% | 0% |
| Currants | *Ribes* spp. | 1900–***1910***–1920 | 1076 | 32 | 35 | 111,189 | 1.1% | 4.6% |
| Cherries | *Prunus avium/P. cerasus* | 1890–***1910***–1930 | 3165 | 19 | 61 | 114,022 | 4.8% | 6.9% |
| Pears | *Pyrus* spp. | 1890–***1910***–1950 | 10,857 | 241 | 1016 | 234,295 | 3.9% | 16.4% |

**Table 1.** *Cont.*

| Agricultural Category and Enterprise | Latin Name (*Genus Species Subspecies*) | Boom/Bust Years (Start–*Peak*–End) | ----Peak Production Year Estimate---- | | | | 2017 Percent of Peak Prod. Year | |
| --- | --- | --- | --- | --- | --- | --- | --- | --- |
| | | | Farms | Area (ha) | Product [1] (metric t) | Real Value (2017 USD) | Product | Value |
| Blackberry | *Rubus* spp. | 1900–***1920***–1945 | 2198 | 114 | 127 | 388,062 | 6% | 7.7% |
| Plum/Prune | *Prunus domestica* | 1900–***1920***–1950 | 6792 | 54 | 264 | 379,093 | 2.2% | 1.2% |
| Apples | *Malus domestica* | 1890–***1920***–1978 | 34,600 | 24,396 | 87,622 | 47,753,831 | 14% | 24.8% |
| Strawberry | *Fragaria* × *ananassa* | 1900–***1940***–1940 | 2168 | 293 | 1518 | 1,946,278 | 31% | 121.9% |
| Raspberry | *Rubus idaeus* | 1900–***1940***–1945 | 1783 | 204 | 179 | 498,673 | 26.4% | 61.7% |
| Peaches | *Prunus persica* | 1900–***1950***–1964 | 1097 | 19 | 139 | 216,309 | 60.8% | 57.9% |
| **TEXTILE/ OTHER** | | | | | | | | |
| Textiles | Silk, flax, & hemp | 1840–***1840***–1910 | 76 | 459 | 35 | n/a | 0% | n/a |
| Hops | *Humulus lupulus* | 1840–***1870***–1900 | n/a | 378 | 135 | 560,045 | 1.4% | 4.3% |

[1] Estimated livestock product/carcass, crop harvest measured in metric tons (t). [2] Horses numbered 109,156 in 1890. [3] Angora goat production as mohair for fiber (not carcass weight) from 168 goats in 1910.

Hay production peaked around 1920 at 1,094,385 t as harvested on 496,582 ha (Table 1). Dry matter (DM) production of total livestock forages and feeds plateaued from 1880 to 1925 (Figures 3a and 5) ranging from 903,100 to 1,014,792 t harvested annually with 2017 production (410,494 t) only 40.5% of 1920's peak production. Since the 1964 Agricultural Census, higher energy corn and sorghum silages, higher protein alfalfa silage, and grass silage (i.e., haylage) have replaced more traditional dry hay such as grass, alfalfa, and small grain hays (Figure 6). Peaks for crops directly fed to livestock on-farm (Table 1, Figure 5) from 1910 to 1950 included (1) 8411 t (920 t DM) of root crops in 1920, (2) 20,724 t (18,444 t DM) of unthreshed feed oats in 1925, (3) 34,336 t (10,301 t DM) for corn hogged, grazed, or cut for fodder in 1935, and (4) 58 t (4.51 t DM) of pumpkins in 1940. Forage seed production peaked around 1860 at 1460 t (Table 1).

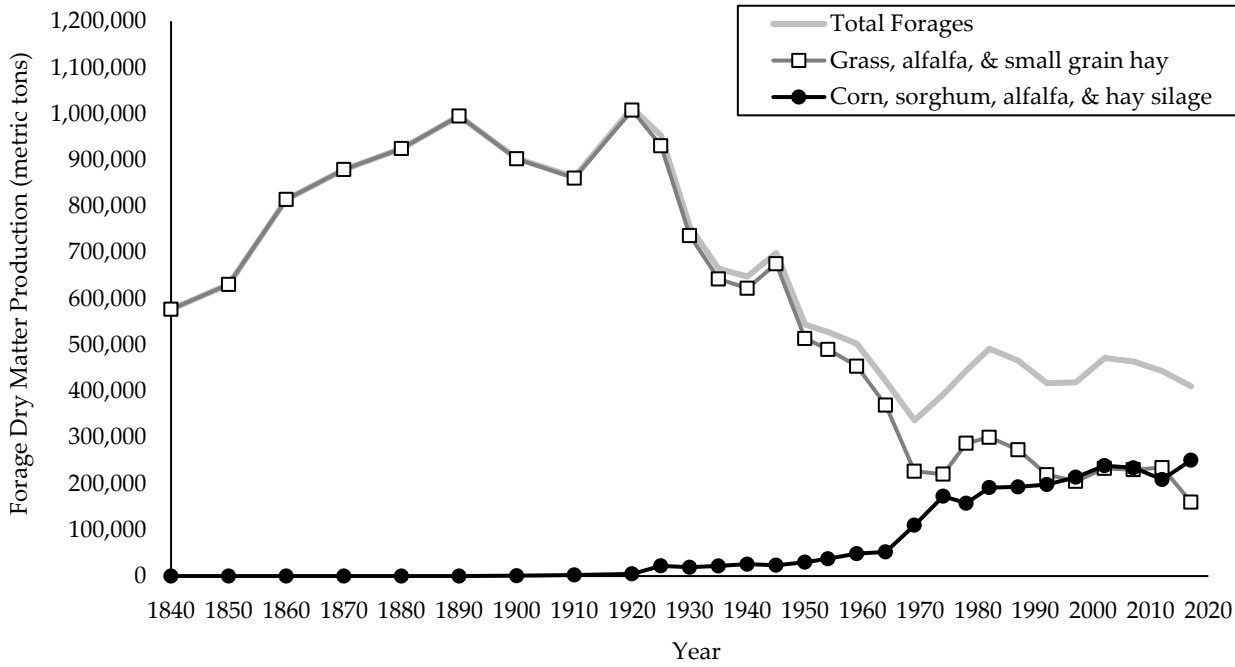

**Figure 6.** Forage dry matter production in metric tons (t) from 1840 to 2017 for Maine, USA.

Traditional livestock peak live weight and production could not be determined for hogs due to a lack of Agricultural Census data prior to 1840 for specific livestock and crops (Table 2). Cattle (beef and dairy) live weight was at its zenith around 1860 at 172,883 t, while 1840 live weights for both sheep (51,043 t) and hogs/pigs (16,506 t) could not be confirmed as peaks (Figure 7a) due to a lack of Agricultural Census data prior to 1840. Horse live weight peaked around 1890 at 46,551 t (Table 2, Figure 7a). Total poultry (broiler and layer chickens, turkeys, ducks, etc.) live weight reached its maximum around 1978 at 184,729 t (Figure 7a).

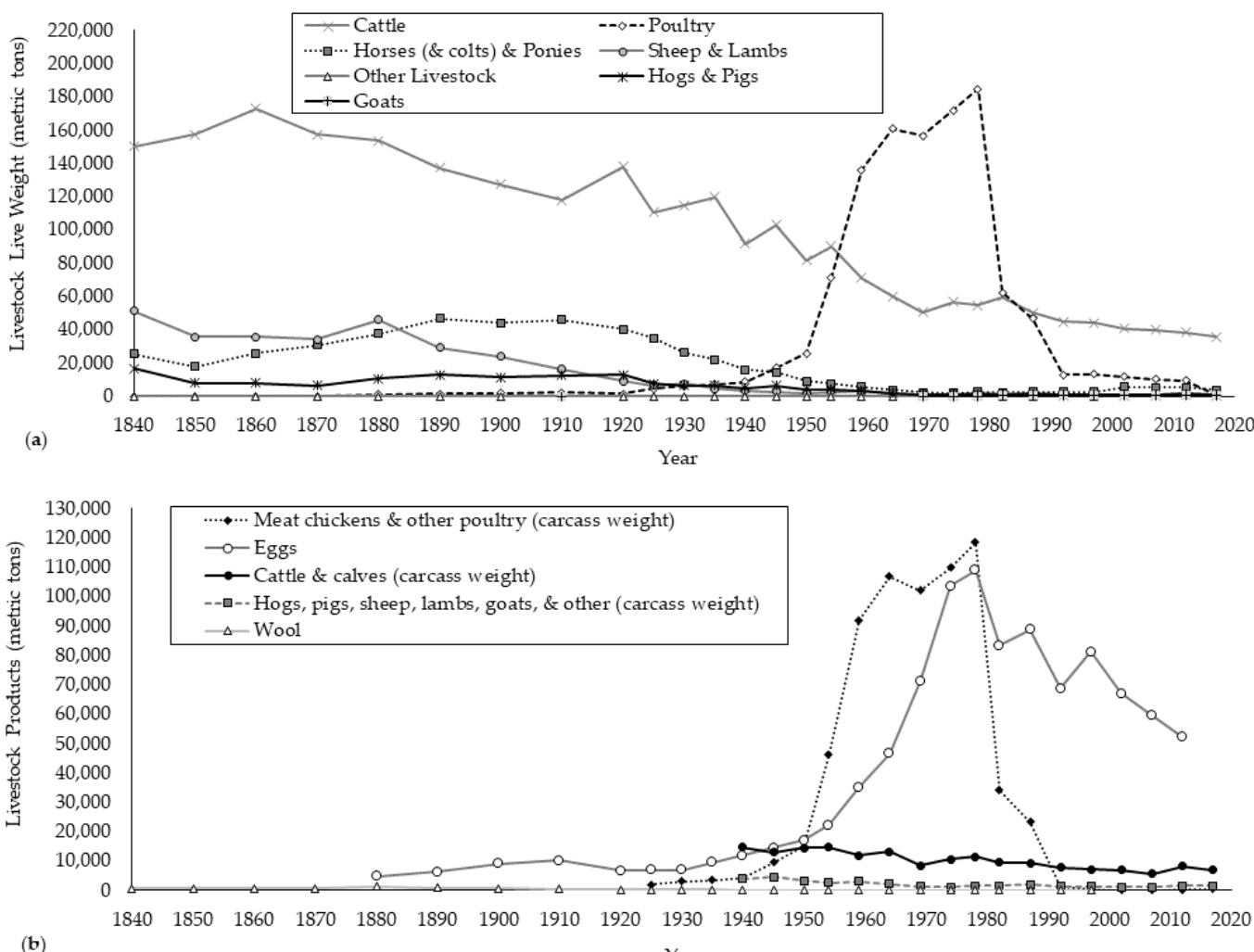

**Figure 7.** (**a**) Livestock live weight and (**b**) livestock product production as carcass weight or product weight in metric tons (t) from 1840 to 2017 for Maine, USA.

**Table 2.** Agricultural booms & busts for more recent specialized/niche systems in Maine, USA, calculated or summarized from publicly available USDA-NASS statistics. Reprinted/adapted with permission from Ref. [24]. 2022, U.S. Department of Agriculture.

| Agricultural Category and Enterprise | Latin Name (*Genus Species Subspecies*) | Boom/Bust Years (Start–*Peak*–End) | - - - -Peak Production Year Estimate- - - - | | | | 2017 Percent of Peak Prod. Year | |
|---|---|---|---|---|---|---|---|---|
| | | | Farms | Area (ha) | Product [1] (metric t) | Real Value (2017 USD) | Product | Value |
| **LIVESTOCK** | | | | | | | | |
| Cattle (dairy) | *Bos taurus* | 1850–***1900***–2017 | 59,299 | - | 409,282 | 482,207,450 [1] | 69.8% | 27.9% |
| Turkeys | *Meleagris gallopavo domestica* | 1930–***1954***–1969 | 567 | - | 1970 | 5,336,560 | 5.0% | 2.6% |
| Broilers | *Gallus gallus domesticus* | 1945–***1978***–1987 | 345 | - | 117,718 | 163,162,608 | 0.3% | 0.03% |
| Eggs (dozen) | *Gallus gallus domesticus* | 1950–***1978***–2012 | 822 | - | 108,958 | 162,158,469 | 47.8% [2] | 40.7% [2] |
| Ducks | *Anas platrhynchos domesticus* | 1974–***1978***–1982 | 74 | - | 689 | 654,121 | 6.5% | 6.5% |
| Rabbits | *Oryctolagus cuniculus* | 1978–***1992***–2002 | 59 | - | 67 | 414,407 | 31.1% | 8.9% |
| Quail | *Coturnix coturnix* | 1992–***1997***–2017 | 4 | - | 0.92 | 5511 | 73% | 125.8% |
| Emus | *Dromaius novaehollandiae* | 2002–***2002***–2012 | 6 | - | 0.94 | 10,076 | 33.3% [2] | 62.8% [2] |
| Angora Goat | *Capra hircus aegagrus* | 2002–***2007***–2017 | 37 [3] | - | 0.81 [3] | 8190 [3] | 30.6% [3] | 8.8% [3] |
| Goats | *Capra hircus* | 1997–***2007***–2017 | 116 | - | 39 | 107,165 | 66.8% | 141% |
| Guinea Fowl | *Numida meleagris* | 2012–***2012***–2017 | 122 | - | 0.44 | 737 | 87.0% | 87.0% |
| **FORAGES** | | | | | | | | |
| Sorghum | *Sorghum bicolor* | 1969–***1974***–1987 | 60 | 667 | 17,328 | 202,036 | 2.9% | 11.9% |
| Corn silage | *Zea mays* | 1964–***1974***–2017 | 805 | 15,122 | 466,462 | 8,226,001 | 83.5% | 227.5% |
| Alfalfa (hay) | *Medicago sativa* | 1954–***1992***–2017 | 774 | 13,995 | 52,437 | 13,815,513 | 37.9% | 30.7% |
| **GRAIN/ OILSEED** | | | | | | | | |
| Oats | *Avena sativa* | 1840–***1910***–2017 | 22,029 | 48,963 | 61,432 | 17,683,787 | 32.0% | 21.5% |
| Sorghum | *Sorghum bicolor* | 1969–***1969***–1978 | 4 | 62 | 142 | 14,953 | 12.9% | 60.2% |
| Wheat | *Triticum aestivum* | 1974–***1974***–1982 / 2007–***2012***–2017 | 59 / 19 | 777 / 968 | 2073 / 2576 | 300,211 / 471,630 | 17.3% / 13.9% | 30.3% / 19.3% |
| Canola | *Brassica napus* | 1997–***2002***–2017 | 20 | 621 | 1131 | 510,770 | 5.1% | 4.3% |
| Barley | *Hordeum vulgare* | 1992–***2002***–2017 | 112 | 10,464 | 39,741 | 4,425,358 | 63.3% | 81.0% |
| Spelt/Emmer | *Triticum spelta/T. turgidum* | 2007–***2007***–2017 | 6 | 18 | 55 | 17,473 | 31.3% | 86.5% |
| Triticale | ×*Triticosecale* spp. | 2007–***2007***–2017 | 4 | 13 | 38 | 16,813 | 25.8% | 38.7% |
| Rye | *Secale cereale* | 2012–***2012***–2017 | 23 | 1687 | 6556 | 968,960 | 31.6% | 38.9% |
| **POTATO + CROPS** | | | | | | | | |
| Potatoes | *Solanum tuberosum* | 1910–***1950***–2017 | 14,904 | 58,028 | 1,791,470 | 313,564,930 | 38.5% | 49.4% |
| Green peas | *Pisum sativum* | 1935–***1964***–1997 | 435 | 4373 | n/a | n/a | - | - |
| Sugar beets | *Beta vulgaris vulgaris* | 1964–***1969***–1974 | 141 | 3840 | 73,064 | 5,559,566 | 0% | 0% |
| Broccoli | *Brassica oleracea* var. *italica* | 1987–***2012***–2017 | 145 | 2555 | n/a | n/a | - | - |
| **VEGETABLES** | | | | | | | | |
| Cucumbers | *Cucumis sativus* | 1900–***1950***–1964 | 982 | 328 | n/a | n/a | - | - |
| Lettuce | *Lactuca sativa* | 1930–***1954***–1969 | 168 | 308 | n/a | n/a | - | - |
| Carrots | *Daucus carota sativus* | 1935–***1954***–1978 | 302 | 172 | n/a | n/a | - | - |
| Dry peas | *Pisum sativum* | 1969–***1969***–1978 | 13 | 250 | 647 | 548,841 | 0% | 0% |
| Dry beans | *Fabaceae* Family spp. | 1969–***1982***–1997 | 115 | 1955 | 1122 | 2,124,836 | 5.7% | 1.8% |
| **FRUITS** | | | | | | | | |
| Grapes | *Vitis vinifera* | 1974–***1982***–2017 | 60 | 15 | 150 | 99,865 | 98.5% | 249.2% |
| Strawberry | *Fragaria* × *ananassa* | 1982–***1987***–2017 | 163 | 238 | 947 | 1,725,185 | 49.6% | 137.5% |
| Blueberry | | | | | | | | |
| Highbush | *Vaccinium corymbosum* | 1940–***1987***–2017 | 110 | 982 | 1118 | 5,288,894 | 30.8% | 31.4% |
| Lowbush | *Vaccinium angustifolium* | 1982–***2014***–2020 | 500 | 9225 | 47,355 | 59,039,649 | 45.4% [2] | 20.9% [2] |
| Cranberry | *Vaccinium oxycoccos* | 1997–***2007***–2017 | 40 | 121 | 841 | 1,457,458 | 30.4% | 11.7% |

[1] Estimated livestock product/carcass, crop harvest (t) and 173,592 dairy cattle in 1900. [2] Egg and emus (2012), lowbush blueberry (2020) used to calculate percent of peak year. [3] Angora goat production as mohair.

Estimated carcass weight for hogs/pigs in 1840 was 10,721 t. Wool production peaked around 1880 at 1259 t (Table 1, Figure 7b) followed by dairy products (milk, cream, cheese, butter) around 1900 at 409,282 t (Table 2). Chicken products exceeded those of other livestock, peaking in 1978 for estimated broiler chicken carcass weight (117,718 t) and eggs (108,958 t) with the decline in egg production more gradual than for broilers since 1978 (Table 2, Figure 7b). Turkey and duck went through shorter boom and bust cycles compared to chicken topping off at estimated carcass weights of 1970 t and 689 t, respectively (Table 2). Production peaks from other specialty livestock ranged from 0.29 t (mohair in 1910) to 67 t (rabbit in 1992) (Tables 1 and 2).

Grains such as wheat, corn, buckwheat, and oats were at their highest and/or peaked between 1840 and 1920 (Tables 1 and 2). While barley and rye were grown during this 19th century time period, more recently their outputs have peaked at 39,741 t in 2002 for barley and 6556 t in 2012 for rye. Wheat has had two minor resurgences, one around 1974 and another around 2012 (Table 2). However, 2017 wheat production was only 1.6% of its historical high in 1840 (Table 1). More recent production cycles have included canola (1131 t in 2002) and non-traditional grains such as spelt/emmer and triticale (Table 2). Dry bean and dry pea production was highest in 1880 (Table 1). Most vegetables and fruits peaked production and/or growing area during the first half of the 20th century (Tables 1 and 2) with potatoes dominating at 1,791,470 t harvested on 58,028 ha in 1950 (Table 2). There were minor rebounds for dry peas (1969) and dry beans (1982). Recent peaks of vegetables/fruits include green peas (1964), grapes (1982), strawberries and highbush blueberries (1987), cranberries (2007), broccoli (2012), and wild blueberries (2014) in Maine (Table 2).

### 3.2. Crop and Livestock Values

Historical peak crop and livestock value of production adjusted for inflation to 2017 U.S. dollars (USD) were highest for dairy products in 1900 (USD 482,207,450), 1950 potatoes (USD 313,564,930), 1920 forage hay (USD 212,351,008), and chicken broilers (USD 163,162,608) and chicken eggs (USD 162,158,469) in 1978. This was followed by 2014 lowbush blueberry (USD 59,039,649), 1920 apples (USD 47,753,831), 1920 oats (USD 17,683,787), 1992 alfalfa hay (USD 13,815,513), 1840 wheat (USD 13,002,123), and 1880 wool (USD 11,931,087). All other crop and livestock products were below USD 10 million. Despite lower production compared to historical peaks, agricultural products with higher values in 2017 were corn silage, quail, meat goats, grapes, and strawberries (Tables 1 and 2). Certain livestock as suggested by their 2017 value have the potential to be future niche species such as tame deer (USD 1,397,000), bison (USD 101,000), pigeons (USD 3547), and peafowl feathers (USD 721). The 2017 crop value of corn harvested as grain is USD 4,859,275 which is 68.2% of the 1850 maximum. Other 2017 values that were historical highs were for alfalfa haylage (USD 4,268,000), peaches (USD 303,117), sweet potatoes (USD 35,084), sunflower seed (USD 3560), and sorghum grain (USD 2187) (Table 3).

### 3.3. Agricultural Diversity Indicators

#### 3.3.1. Richness, Effective Number of Species, and Relative Abundance

Crop/livestock category species richness ($R$ = number of species) increased very modestly for livestock forage/feed, floriculture/propagation/seeds/X-Mas trees, and potatoes and annual crops rotated with potatoes. For mixed vegetables, fruits, nuts, and specialty crops, $R$ increased from 33 to 60 (1900 to 1950), bottomed out to 31 to 34 (1954 to 1969), spiked from 42 to 61 (1974 to 1978), and then bottomed out again to 36 (1982 to 1987), and then increased substantially from 46 to 82 (1992 to 2017). Livestock $R$ increased from 5 to 26 from 1850 to 2012 (Figure 8a).

**Table 3.** Agricultural systems with increasing production potential in Maine, USA, calculated or summarized from publicly available USDA-NASS statistics. Reprinted/adapted with permission from Ref. [24]. 2022, U.S. Department of Agriculture.

| Agricultural Category and Enterprise | Latin Name (*Genus Species Subspecies*) | Growth Years (*Peak–Start–Recent*) | ------Recent Year Estimate------ | | | | 2017 Percent of Peak Prod. Year | |
|---|---|---|---|---|---|---|---|---|
| | | | Farms | Area (ha) | Product [1] (Metric Tons) | Real Val. (2017 USD) | Product | Value |
| **LIVESTOCK** | | | | | | | | |
| Pheasants | *Phasianus colchicus* | *None*–1992–2012 | 6 | - | 7 | n/a | n/a | n/a |
| Bison | *Bison* spp. | *None*–2002–2017 | 12 | - | 13 | 101,000 | 100% | 100% |
| Deer (tame) | *Odocoileus virginianus* | *None*–2002–2017 | 37 | - | 73 | 1,397,000 | 100% | 100% |
| Pigeons | *Columba livia domestica* | *1910*–1992–2017 | 15 | - | 0.17 | 3547 | 43.3% | 43.3% |
| Peafowl [2] | *Pavo* & *Afropavo* spp. | *None*–1910; 2012–2017 | 25 | - | n/a | 721 [2] | - | - |
| **FORAGES** | | | | | | | | |
| Alfalfa (haylage) | *Medicago sativa* | *None*–2002–2017 | 139 | 5750 | 87,400 | 4,268,000 | 100% | 100% |
| **GRAIN/OILSEED** | | | | | | | | |
| Sorghum | *Sorghum bicolor* | *None*–2012–2017 | 3 | 8 | 142 | 2187 | 12.9% | 12.9% |
| Sunflower seed | *Helianthus annuus* | *None*–1997–2017 | 1 | 9 | 9 | 3560 | 100% | 100% |
| Corn | *Zea mays* | *1850*–1974–2017 | 82 | 2929 | 30,327 | 4,859,275 | 68.2% | 68.2% |
| **VEGETABLES** | | | | | | | | |
| Sweet potato | *Ipomoea batatas* | *None*–2002–2017 | 34 | 4 | 9 | 35,084 | 100% | 100% |
| **FRUITS** | | | | | | | | |
| Peaches | *Prunus persica* | *1950*–1982–2017 | 118 | 18 | 84 | 303,117 | 60.8% | 60.8% |

[1] Estimated livestock product/carcass, crop harvest (t). [2] Peafowl numbered 6 in 2017 valued at USD 115.64 per fowl so cannot distinguish if feathers or live animals sold.

Effective number of species (*ENS*) was similar to *R* for potatoes and annual crops rotated with potatoes as well as livestock forage/feed (Figure 8b). The *ENS* for floriculture, propagation, seeds, and X-Mas trees declined abruptly after 1964 (Figure 8b) whereas *R* fluctuated over time with a more recent increase (Figure 8a). *ENS* for the mixed vegetables, fruits, nuts, and specialty crops category had much less pronounced valleys and peak after 1950 compared to *R*. Unlike *R*, *ENS* for mixed vegetables, fruits, nuts, and specialty crops in 2017 did not exceed the 1950 peak (Figure 8).

Area of livestock forage/feed and potato rotation systems have been more relatively abundant compared to the other two crop categories of mixed vegetables, fruits, nuts, and specialty crops as well as floriculture, propagation, seeds, and X-Mas trees. The relative abundance of the livestock forage/feed category has increased, while the relative abundance of the potatoes and crops rotated with potatoes category has increased since 1969 (Figure 9a). Relative abundance of livestock product weight was dominated by broiler chickens and eggs with more recent balance relative to beef and pork (Figure 9b).

### 3.3.2. Diversity Indexes

Crop and livestock category diversity indexes were consistent with calculated effective number of species. For mixed vegetable, fruit, nut, and specialty crops, the Species Diversity Index (*SDI*), Shannon-Weiner Index (*SWI*), and Composite Entropy Index (*CEI*) were similar, while Evenness (*E*) gradually increased over time (Figure 10). For livestock, *SDI*, *SWI*, and *CEI* increased over time with volatility between the years 1930 to 2000, while livestock *E* declined (Figure 11a). There were increases in evenness and diversity in potato systems since the 1970's (Figure 11b) and livestock forage/feed from 1954 to 2007 (Figure 12a). Diversity indexes and *E* for floriculture, propagation, seeds, and X-Mas trees have been more volatile trending upward and downward, respectively (Figure 12b).

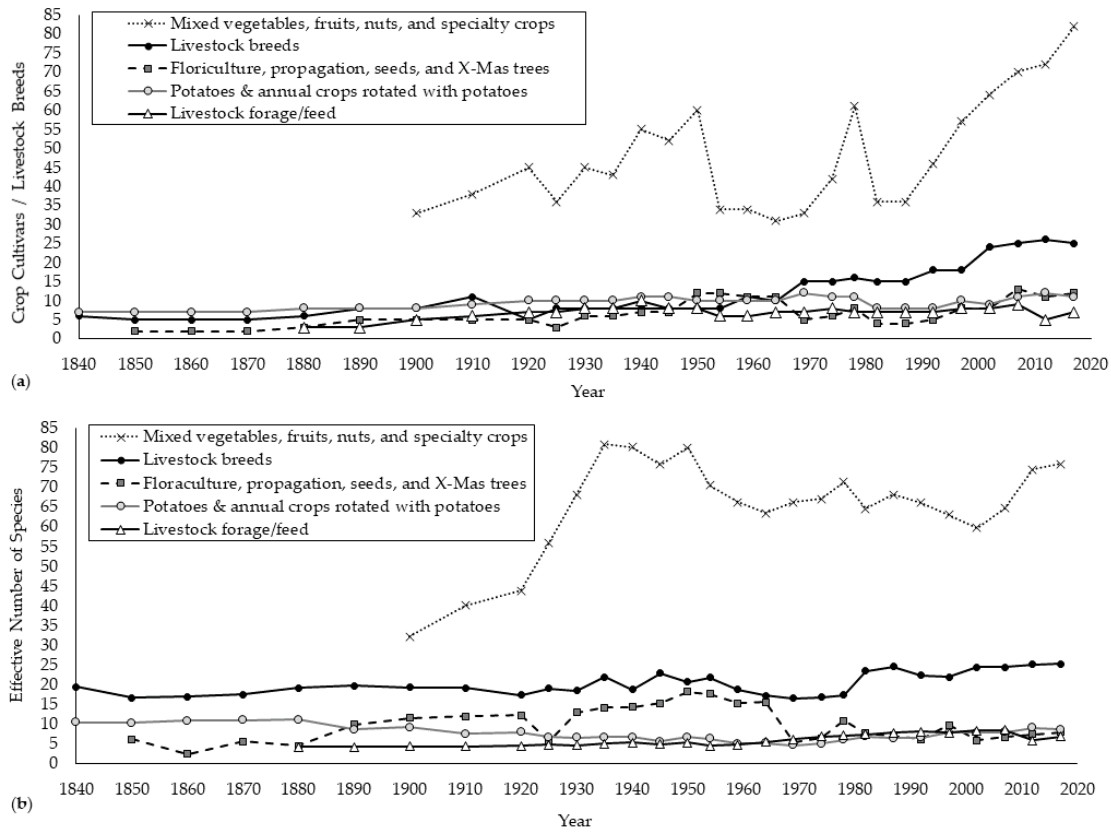

**Figure 8.** Crop cultivar and livestock breed (**a**) richness and (**b**) effective number of species in general categories from 1840 to 2017 for Maine, USA.

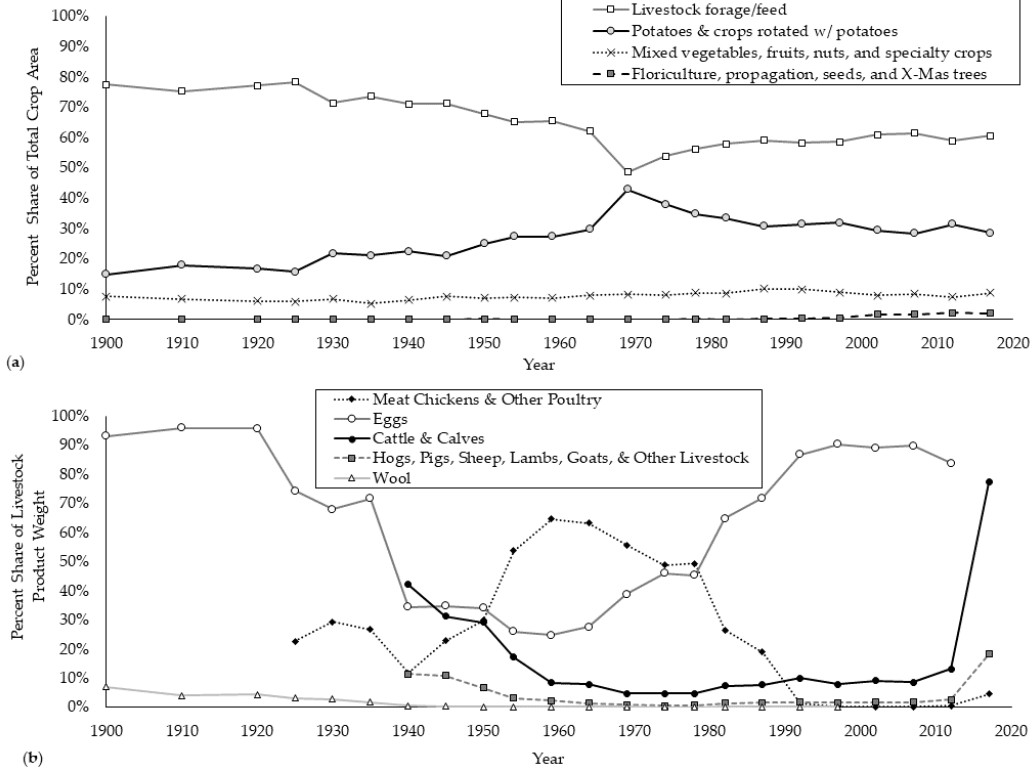

**Figure 9.** Relative abundance of (**a**) crop and (**b**) livestock product categories' percent share of total crop area from 1900 to 2017 for Maine, USA.

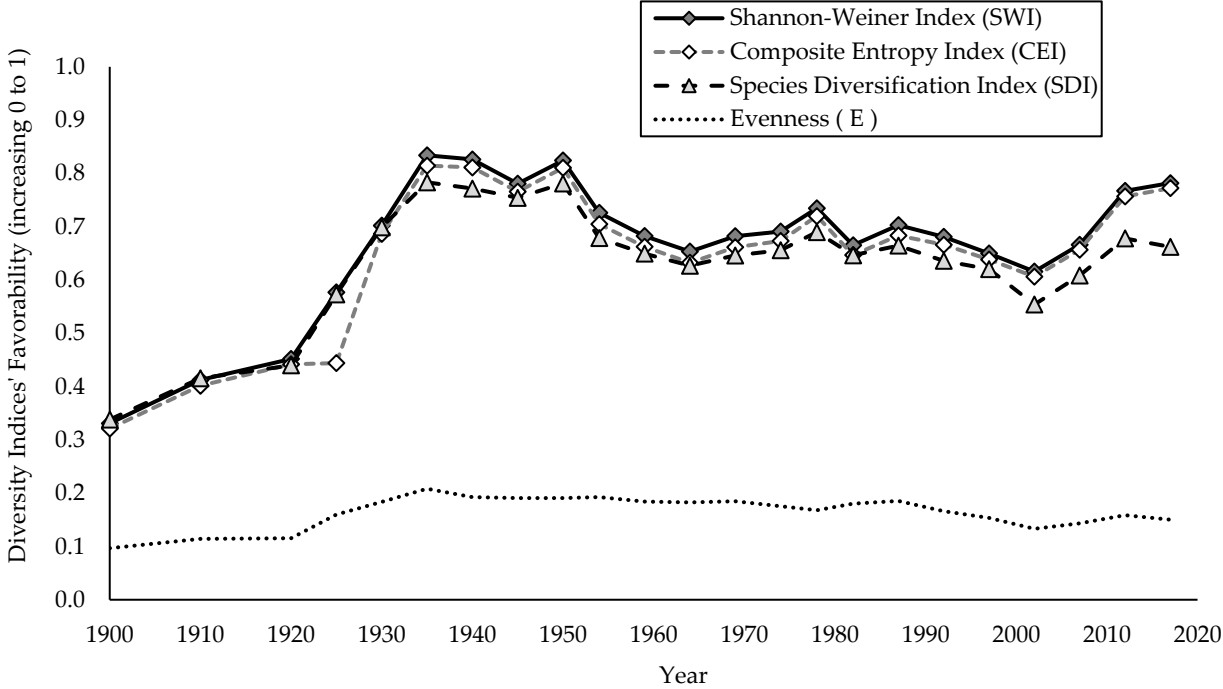

**Figure 10.** Diversity indexes for mixed vegetable, fruit, nut, and specialty crops from 1900 to 2017 for Maine, USA.

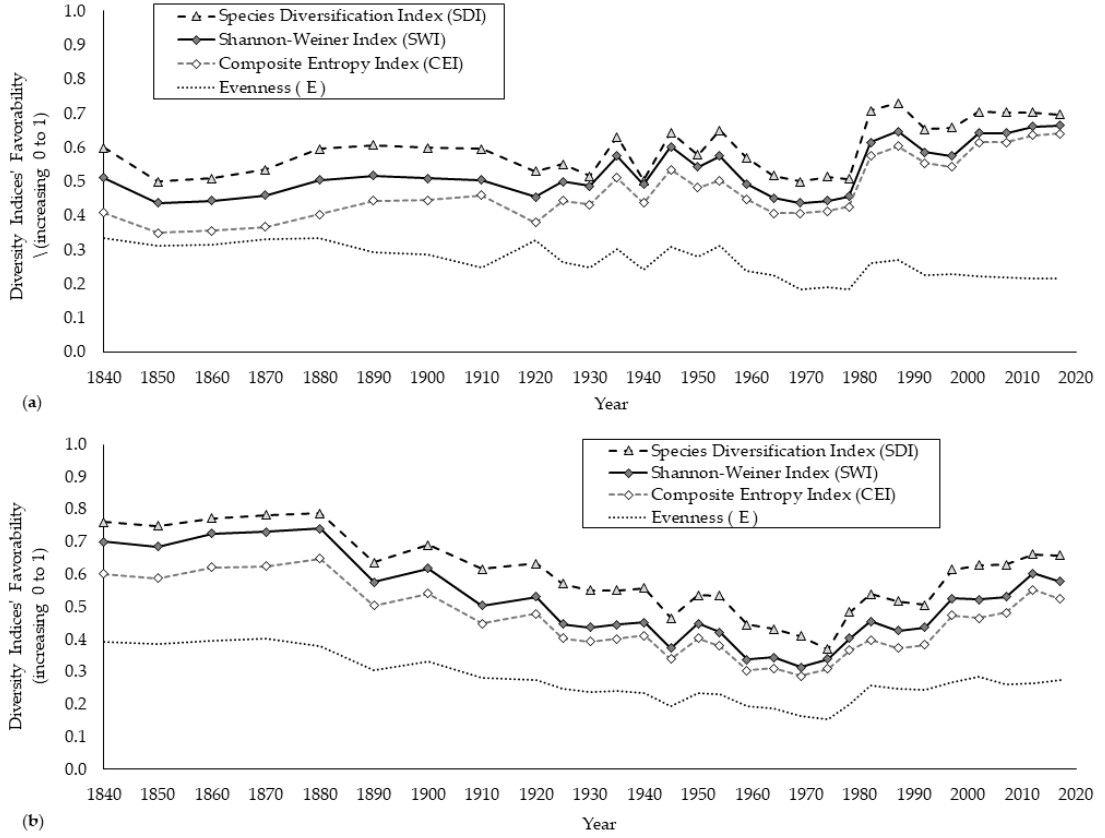

**Figure 11.** Diversity indexes for (**a**) livestock and (**b**) for potatoes and crops rotated with potatoes from 1840 to 2017 for Maine, USA.

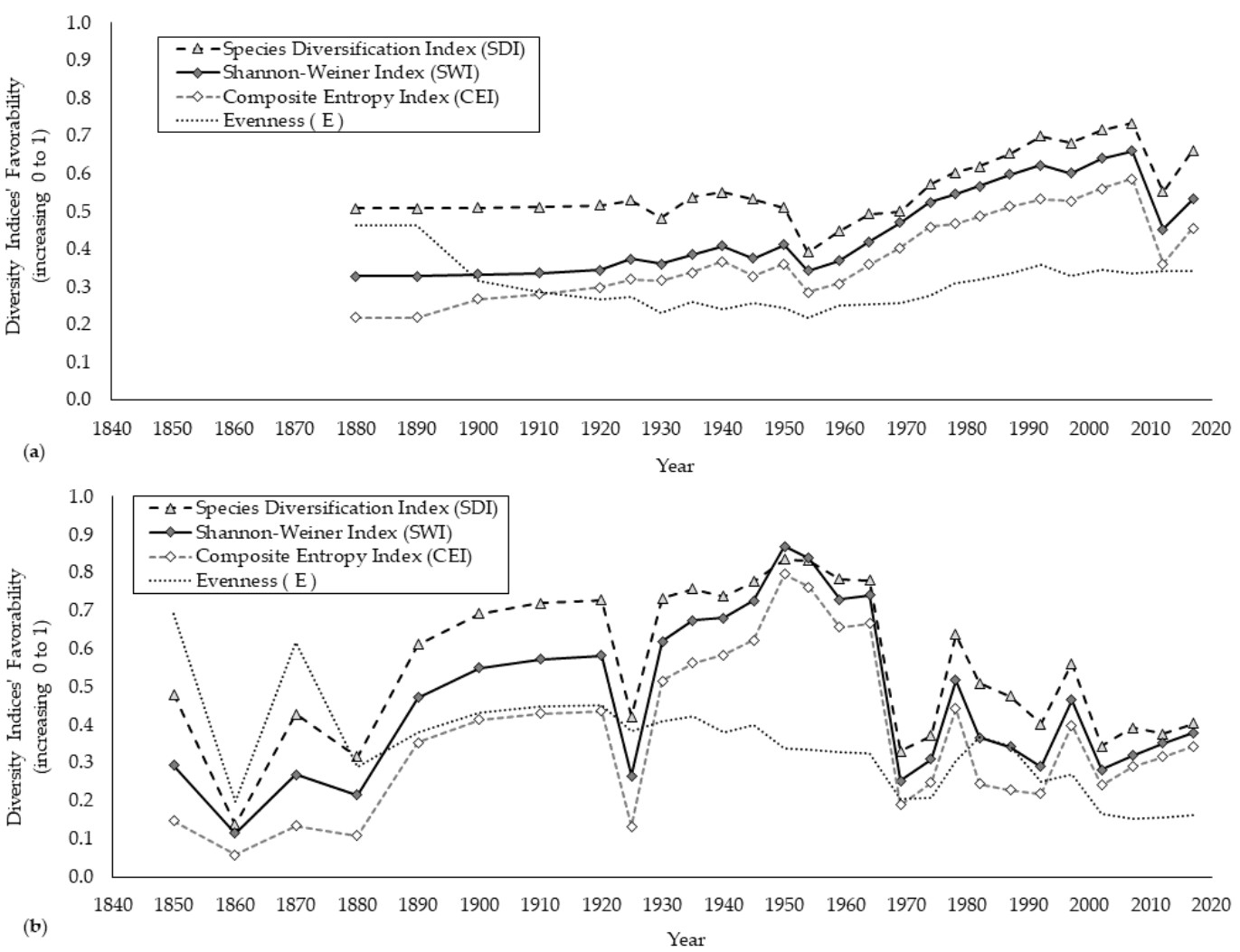

**Figure 12.** Diversity indexes for (**a**) livestock forage/feed and (**b**) floriculture, propagation, seeds, and X-Mas trees from 1880 to 2017 for Maine, USA.

## 4. Discussion and Conclusions

### 4.1. Comparisons and Contrasts to Prior Studies

Compared to past research measuring effective number of species of agricultural crops across the USA, results for Maine (1840 to 2017) were both consistent and different. Maine crops rotated with potatoes (Figure 13a) which are predominantly small grains such as oats (Figure 13b) had crop area following similar trends compared to a state-level USA study from 1870 to 2012 for 22 major field crops [9]. The effective number of species (*ENS*) for these 22 crops ranged between 1 and 7 and peaked during the 1940's and 1960's [9]. For Maine, *ENS* decreased from 10.1 to 3.7 from 1880 to 1969 and then rebounded to 8 in 2012 for crops rotated with potatoes (minus potatoes) for Maine (Figure 8b). A study using USA county-level data for all crop species from 1978 to 2012 found that average *ENS* increased from 5.85 in 1978 to 6.6 in 1997 and then decreased to 5.49 in 2012 for the Northern Crescent region (Great Lakes states, New York, and New England) in the USA [8]. A more recent Geographic Information Systems study analyzing USDA's Cropland Data Layer (CDL at 30 m resolution) of crop categories from 2008 to 2018 found increasing *ENS* in potato producing regions (e.g., northern Aroostook County) and decreasing *ENS* in other areas of Maine [10]. This is consistent with results for Maine's recent *ENS* trends for livestock forage/feed and small grain crops in rotation with potatoes (Figure 8b) as previously discussed.

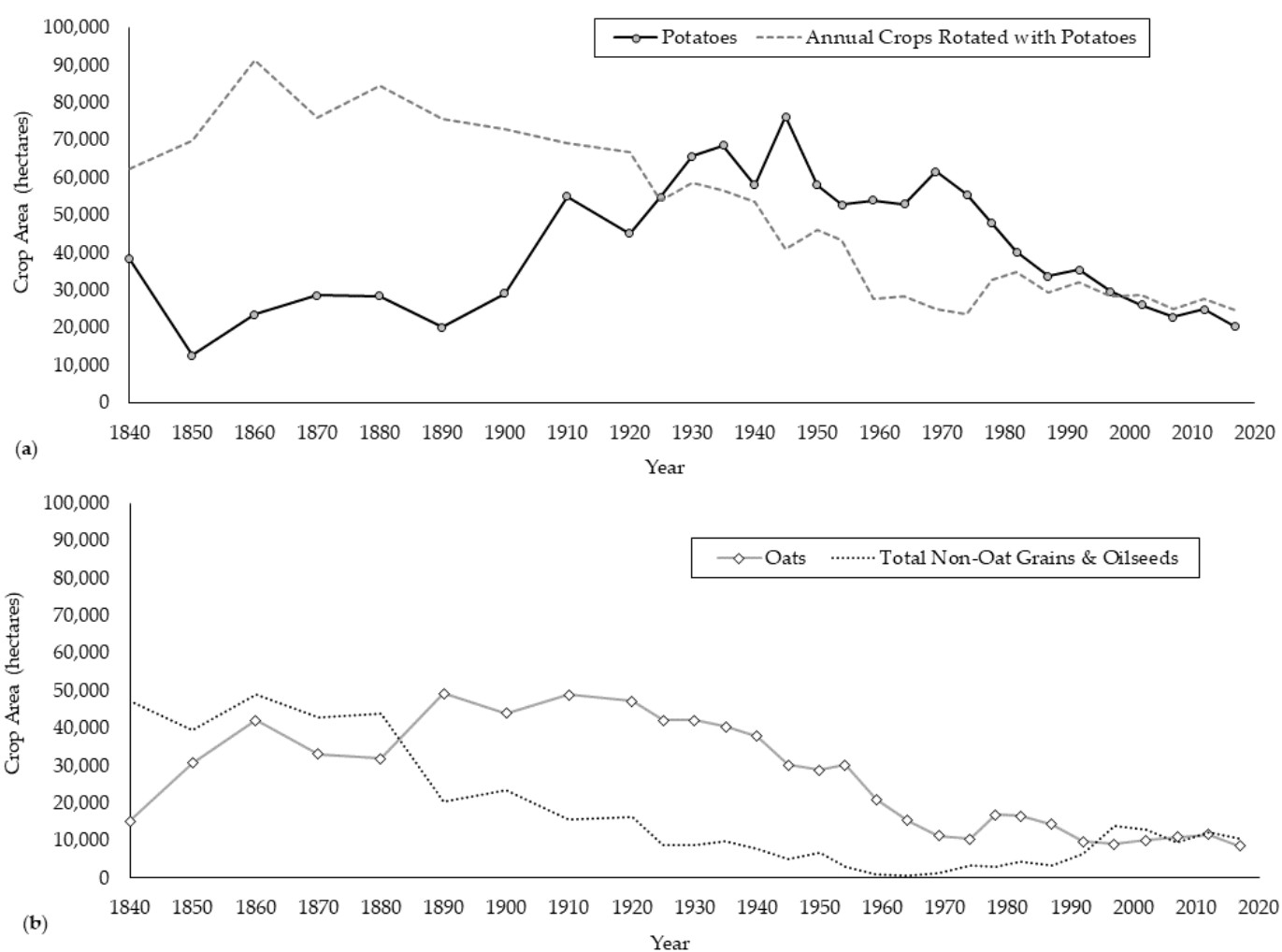

**Figure 13.** (**a**) Potatoes and annual crops rotated with potatoes (hectares) and (**b**) oats and non-oat grains and oilseeds (hectares) from 1840 to 2017 for Maine, USA.

Unlike other states in this region, *ENS* in Maine has increased and not decreased since 2002 for crop species with the exception of livestock forage/feed (Figure 8b) which had a pattern similar to the Northern Crescent region [8]. Maine's recent increase in crop and livestock diversification over the past 20 years has involved mixed vegetables and specialty crops as well as non-traditional livestock (Figures 8, 10 and 11a; Table 3), which have been more difficult to measure by past research on regional trends [8,9]. Additionally, the Geographic Information Systems (GIS) Cropland Data Layer (CDL) data used in [10] is too coarse to distinguish smaller diversified mixed vegetable farms and crop area which have increased in Maine over this time. From 2007 to 2017, the average number of farms growing any one of 55 categories of mixed vegetables increased from 69 to 169 corresponding to an increase from 25.5 to 36.25 average hectares per category over this time [24]. Thus Maine's *ENS* for mixed vegetables, fruits, nuts, and specialty crops increased 27% from 2002 (59.7) to 2017 (75.8) and did not decrease (Figure 8b).

On diversified produce farms in Maine, blocks or beds of mixed vegetable species can be as small as ~10 square meters (m$^2$) with the average sized vegetable farm in Maine being 1.42 hectares (ha) [62]. USDA's Cropland Data Layer GIS pixel size is 30 m × 30 m = 900 m$^2$ = 0.09 ha which could distinguish larger contiguous planting of vegetable species. However, past GIS diversification analysis aggregated grids to a 4 km × 4 km = 16 km$^2$ = 1600 ha [10] which distinguished consistent patterns for regions within Maine, but not at a farm-level or diversified crop-specific scale. Clearly there is a need for finer scale

evaluations of crop/livestock species diversity from analyses of USDA Agricultural Census and Survey data or similar types of national statistics to complement coarser scale analyses.

*4.2. Historical Determinants of Agricultural Specialization in Maine*

Maine has witnessed boom and bust in both ruminant livestock and horses (Figure 4) as well as the hay, forages, and other livestock feeds (Figure 5) fed to these livestock. During the early 1800's in Maine, livestock were fed manually harvested and cured hay, highly subject to reductions in yield and quality from adverse weather conditions throughout the year [63]. Land in Maine from 1850 to 1910 used to be ~30–70% cleared for farmland as farms were dependent on horses in this pre-tractor era [6]. Maine's sheep industry peak (Figure 7) was driven by the doubling of wool prices during the Civil War [64]. Declines in livestock (Figure 7) and crops (Figure 5) during the mid- to late-1800's can also be explained by the end of the family farming era in Maine as many younger farmers moved to take advantage of Ohio's cheap yet more productive agricultural land [65].

While Maine's boom and bust for ruminant livestock and horses characterized the 19th century, the boom and bust of the potato industry in northern Aroostook County, Maine, dominated the 20th century. Aroostook County farmers were originally diversified and dependent on logging as these two industries were tied together until the 1870's to 1890's. Farmers had to work in the woods to make ends meet. It was not until railroad access was finished in the 1870's that farmers were able to specialize into potatoes by securing more reliable out-of-state markets such as that for potato starch [66,67]. Self-sufficiency in farming in the mid-1800's gave way to industrial agriculture. Additionally, contributing to the specialization boom for Maine potatoes were the establishment of the Maine Agricultural Experiment Station in 1915, local agricultural societies, fairs, and clubs as well as the Grange [67]. This followed an ivory silo issue in 1870's and 1880's where farmers were very distrustful of universities and education was biased against teaching the practice of farming instead focusing on the theory and current research of the academic disciplines related to farming [68].

A common theme behind the decline in Aroostook County's specialized potato production is The County's distance and isolation from both job and product markets. Agriculture in Maine since the early 1900's has been far away from East Coast, USA, markets compared to the rest of New England. There has also been historical brain drain of younger people seeking careers out-of-state which has made economic let alone agricultural stability more challenging. A key difference is that in 1930, more people were involved in farming and self-sufficient food production and procurement. Since the 1980's, producers have been responsible for feeding more people per farm so there is more pressure to maintain farm solvency through farm specialization versus being able to rely on other enterprises and activities to maintain the viability of the farm household [69].

Maine's potato rotations were longer prior to the early 1900's with potato production during the boom intensifying so that potato area exceeded commodity crops commonly rotated with potatoes from 1925 to 1997 (Figure 13a). The dominant crop rotated with potatoes was oats from 1890 to 1992 with more balance of potato rotation crops bookended before and after this one hundred year period (Figure 13b). Specialized agriculture capitalizing on economies of scale can result in reduced farm resilience over time especially if there is a lack of strong centralized marketing [21]. In Maine, this was exemplified by the failure to establish sugar beets as a complementary commodity rotation crop in potato rotations during the late 1960's and early 1970's (Figure 5). Agricultural industry specialization combined with Aroostook County's isolation has made recent adoption of sustainable systems involving crop-livestock integration (CLI) more challenging [70], even though there are mutual economic benefits for specialized potato and dairy farmers to integrate their cropping systems [71]. There is also a lack of regional CLI infrastructure [72] that could facilitate cost-effective movement of excess manure from larger livestock farms in central/southern Maine to northern Maine's non-integrated potato farms.

Crop boom and busts for vegetables primary grown for canning included peaks of 6654 hectares (ha) of sweet corn in 1930, 1616 ha of green beans in 1945 (Table 1), and 4373 ha of peas in 1964 (Table 2) [24]. During the latter half of the 20th century (1940 to 1985), Maine agriculture was characterized by the Green Revolution treadmill of getting bigger or getting out juxtaposed against a focus on diversification into activities not widespread at the time such as sheep production in addition to direct marketing. Specialized commodities in Maine included potatoes, dairy products, broilers, eggs, apples, wild blueberries, and cattle which made up 84% of the value of farm production in 1974 [73].

Agricultural commodity specialization, booms, and busts can be driven by global and regional factors such as trade and/or competition between nations or regions [74]. Commodity production commonly clusters around adequate agricultural support industries as well as greater availability of key agricultural inputs such as labor [75]. Legal factors (e.g., environmental regulations) can also shift entire agricultural industries. For example, the collapse of Maine's broiler chicken industry by 1992 (Figure 5) and the rapid, subsequent shift in broiler production from Maine to the Southeast USA illustrates hard to counteract national/regional forces and policies. Lower labor costs and less stringent regulations in the Southeast USA relative to Maine shifted the broiler chicken industry to this region by the late 1980's and early 1990's [18,73].

### 4.3. Recent and Future Diversification Directions

Maine's recent crop/livestock diversifications since the 1970's has flourished in the wake of the collapse of the traditional and conventional agricultural systems previously discussed. Rather than crop land being ~30–70% of Maine's land area from 1850 to 1910, by 1970, Maine's farmland had declined to only ~10% of total land area [6]. Despite a more limited agricultural land base compared to historical periods (Figure 2), Maine's agriculture has shifted to more diversified, smaller farms [76]. There are three areas where Maine can continue to diversify its agricultural systems: (1) growing more livestock feed in-state rather than importing this from Canada and/or the Midwest USA, (2) mixed vegetables, fruits, nuts, and specialty crops, and (3) crops rotated with commodity potatoes.

The early to mid-1900's showcased diverse livestock feed (Figure 5) such as unthreshed oats bound for feeding, corn hogged in field, as well as forage turnips and pumpkins [24]. Given increasing effective number of species (Figure 8b) and livestock diversification (Figure 11a) in Maine over the past couple of decades, re-adopting both harvested and in-field supplemental livestock feed presents a tremendous growth opportunity and a future pathway to livestock feed self-sufficiency using cover crops [77]. Livestock forage diversification research has evaluated integrating non-traditional forages into Maine's organic dairy farm systems such as triticale (*Triticosecale rimpaui* Wittm.) and brown midrib sorghum-sudangrass (*Sorghum sudanense* (Piper) Stapf) [25]. Other recent initiatives to diversify organic dairy farm crop rotations have included integrating wheat (*Triticum aestivum*), soybeans (*Glycine max* L.) [78], and sunflower [24] meal as a by-product of sunflower oil production (personal correspondence, Richard Kersbergen, University of Maine). There has been a lack of focus on forage crops consumed in-field as was prevalent in the early to mid-1900's. This presents an opportunity to go "Back to the Future" to diversify crop rotations on farms with non-confined livestock such as hogs, beef cattle, and organic dairy herds, while simultaneously reducing reliance on imported feeds.

Diversification indicators for Maine's mixed vegetable, fruits, nuts, and specialty crops category from 1840 to 1970 support the theory that farms initially become more diverse as market size increases, but then once a critical threshold is reached, farms become increasingly less diverse and more specialized where diversification indicators follow a reverse U-shape over time [79]. Unlike other regions in the USA, Maine mixed vegetables and specialty crops have recently become more diverse since 1970, not increasingly less diverse. Similar to West Bengal, India from 1970 to 2005, demonstrating diversification is influenced by smaller farms and growth of infrastructure networks [80], the Maine Organic Farmers and Gardeners Association has been instrumental in supporting smaller organic

farms since 1971 [81]. This recent shift to local and regional food systems presents an opportunity to diversify farming in Aroostook County, which has the greatest potential in New England, USA, for produce distribution [82].

Maine, USA, has had two periods of increased mixed vegetable, fruit, nuts, and specialty/other crop diversification since the 1950's, (1) the back-to-the-land movement of the mid-1970's [83] and (2) the local food movement over the past 15 years. Diversified producers in Maine have focused on economies of scope, retaining a greater share of consumer expenditures [84]. One way to do this is to produce and direct market higher value crops such as sweet potatoes (Table 1), strawberries, and grapes (Tables 1 and 2). In Maine, the 2017 value for these three crops was proportionally greater relative to their inflation-adjusted historical peak values. Non-traditional crops can capitalize on early entry into the market but many of these crops require season extension (e.g., sweet potatoes, ginger, etc.). Profits could also be eroded by competitive entry from other farmers as the market for these non-traditional crops becomes more saturated or by increases in home gardening. For example, 7 out of 54 mixed vegetable/field fruit crops had production declines from 2012 to 2017 (tomatoes, peppers, turnip greens, sweet corn, cucumbers, green beans, and broccoli) with >20% drops in area for tomatoes (−54.4%), peppers (−38.2%), and turnip greens (−26.5%) [24].

Maine's potato rotations were historically more integrated with livestock forages [85]. Although there has been increased diversification in potatoes and potato rotation crops (Figure 11b) since the 1969 boom and bust in sugar beets (Figure 5), recent potato rotation crops have been dominated by commodities such as barley grown for malting with limited area devoted to higher value small grains such as wheat [24]. Barley recently peaked at 10,464 ha in 2002 compared to only a 2012 peak of 968 ha for wheat (Table 2) despite more recent interest and research on expanding organic wheat production in Maine and Vermont, USA [86]. With almost 90% of Maine's potato production concentrated in Aroostook County in the northeast corner of the state [85], there have been limited options for higher value, more profitable potato rotation crops such as broccoli [87]. Recent diversification into broccoli production in Aroostook County has peaked at 2555 ha in 2012 (Table 2). Future efforts could focus on other higher value commodity vegetables for produce or processing that can be rotated with potatoes.

Future research could also statistically test potential drivers of recent diversification trends in Maine. These potential drivers include socio-demographic characteristics of diversified farmers including off-farm income stability [59,88], species selection and input use [61], agricultural technologies such as irrigation and equipment [10,61,88], regional infrastructure [88], population density [59], and access to Extension, market information, and rural credit [88]. Future studies could also evaluate potential food security benefits of crop-livestock integration [89] in addition to better quantifying diversification and benefits from inter-cropping [90]. Aggregate crop data and agricultural statistical surveys do not measure if crops are inter-cropped so cultivar/species richness may underestimate positive synergistic impacts.

**Funding:** This research received no external funding.

**Institutional Review Board Statement:** Not applicable.

**Informed Consent Statement:** Not applicable.

**Data Availability Statement:** Publicly available datasets were analyzed in this study. These data can be found here: U.S. Department of Agriculture (USDA) Cropland Data Layer [https://nassgeodata.gmu.edu/CropScape/] accessed on 21 October 2022 and USDA Agricultural Census and Survey [https://quickstats.nass.usda.gov/] accessed on 21 October 2022.

**Acknowledgments:** The author would like to thank MDPI Sustainability for the opportunity to serve as guest editor for the Special Issue "Sustainable Agricultural Development Economics and Policy". Thanks also to all the researchers who have taken the time to continuously improve their manuscripts for this Special Issue. Many thanks to Ronaldo A. de Oliveira at AgriSciences, Universidade Federal de Mato Grosso, Sinop, Mato Grosso state, Brazil, for creating the Maine crops map using U.S. Department of Agriculture's (USDA's) Cropland Data Layer as recommended by Tara King at USDA Natural Resource Conservation Service in Bangor, Maine. Nancy McBrady and Willie Sawyer Grenier at Maine Department of Agriculture (MDA) authorized permission to use MDA's map of major crop growing areas in Maine. Thanks also to John Harker, Richard Judd at the University of Maine, and David Vail at Bowdoin College for advising on map research. Also, thanks to Jim Barrett at USDA National Agricultural Statistics Service. Thanks to Tim Schermerhorn at U.S. Bureau of Labor Statistics for providing historical Producer Price Index data. The organization and writing of this research was substantially improved with edits and feedback from three anonymous reviewers.

**Conflicts of Interest:** The author declares no conflict of interest. Supporting entities had no role in the design of the study; in the collection, analyses, or interpretation of data; in the writing of the manuscript, or in the decision to publish the results.

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
