# Peer review of "Back to the Future: Agricultural Booms, Busts, and Diversification in Maine, USA, 1840–2017"

_sustainability, doi:10.3390/su142315907_

Round 1
Reviewer 1 Report
You have to revise the article based on my comments

Author Response
Reviewer 1
Sustainability Agriculture are major issues in some countries. The manuscript is clear, and relevant in the field of sustainability, the article already answer the objective. The data suitable enough to answer the article objectives. The authors already explain well the Agricultural booms, busts, and diversification in Maine based on perspective of sustainable development. The article is quite interesting, the topic is related with the journal scope. The references cited are relevant, but the authors need add some (2-3 related references again). The author wrote the introduction well and supported with new references. The references use also related with article. The methodology also is completed written. The study design is appropriate, to test the hypothesis. The sample used in good number, and the analysis used in the methodology is suitable to test the hypothesis. The results is clear, the table and graph is well presented and easy to understand by reader. The author success to explain the Crop and Livestock values with the very appropriate data, the data start from 1840-2020, enough to represent the actual condition. The author success to explain mixed vegetables, fruits, nuts, and specialty crops 379 in 2017. In the sufficiency data the data presented is suitable enough. The authors also success presented the Crop and livestock category diversity indexes with the appropriate data. In Figure 6 the authors success presented the Crop cultivar and livestock breed richness in general categories from 1840 to 2017 for 392 Maine, USA. The authors just need make the result more related each other. The conclusions already well written.
The revisions from other reviewers has re-focused the manuscript. The manuscript results have been focused better in relation to each other and about 13 cited references have been deleted. However, we agree with your suggestion to add more references and the 7 we have added improve the depth of the discussion/conclusions.
For example, 3 of the 7 added citations focus on why agricultural specialization and diversification occur and are:
- Gurgel et al. 2021 – Added as citation #73
- Carpenter et al. 2022 – Added as citation #74
- Smith 2004 (book chapter) – Added as citation #75
The improvements to the discussion/conclusions section have tied research results more closely together and made results more related to each other.
Reviewer 2 Report
I thank the Publisher for sending me this work, which demonstrates extensive knowledge of the sector and of the data series, being a very interesting proposal. However, the work suffers from a number of important issues:
1. The number of keywords is excessive, it must focus the interest of the article.
2. It has structural deficiencies:
• The abstract does not follow the indications of the instructions for authors (https://www.mdpi.com/journal/sustainability/instructions), its reading must clearly indicate the content of the work.
• The introduction is disorganized, it is the result of the juxtaposition of elements, without the different paragraphs being related to each other. You have to follow a clear structure (build a speech) that goes from the general to the specific. For example (not necessarily, the author can search for another order that he considers to have scientific potential): a) Forests and crops in temperate areas; b) agricultural booms and crises (it is necessary to elaborate on their causes, without forgetting the thematic axis: Sustainable Agricultural Development Economics and Policy); c) North America/United States; d) Maine (this could be another heading or appear as a study area in the Methodology). It must have a structure according to results and discussion/conclusion.
• The results must coincide in the structure with what is stated in the methodology and the introduction (correlation of the parts).
3. Deficiencies in the methodology: it is clear from the results that the author masters the methodology, but it is not properly explained. It should not start from what the reader knows about the subject, he should present it in the following way: "we take this data and apply these methods". On the other hand, the subheadings must coincide with what is done (analysis of Agricultural Booms and Busts…). I should comment on the sources he uses (and the methodologies used), which he only cites.
4. Content deficiencies:
• Although the introduction addresses the causes of the changes that occur, the discussion/conclusions are excessively descriptive (comparative) and do not go into the reasons why the changes occur (international competition, trade agreements, etc.). This must be solved by giving a new approach to the introduction (focused on analysis methodologies), or by introducing causality in the discussion/conclusions.
• You should avoid decontextualized quotes, which do not coincide with the object of study or with the context. Talking about the boom and crisis of production (lines 46-53) in the European Neolithic to introduce a capitalist economic cycle is inappropriate, for several reasons: a) it is based on specific studies carried out with archaeological techniques (obviously different sources); b) the modern and contemporary production crises (the potato crisis in Ireland or the great famine...) are documented (even some of them, statistically) and coincide (not in causes, but in development) with the cycles of which the author speaks; c) there are archaeological studies on pre-colonial America, which may be more related to the work (but do not). On the other hand, there must be a temporal continuity, we cannot jump from the Neolithic to the date for which we have data.
5. Question of sources: in the discussion/conclusion sources are forced to make comparisons, which do not coincide neither in time nor in context with the results obtained. If we talk about the production crisis in Maine, we must contextualize it in a similar production crisis, not in data obtained from elsewhere in different contexts (India, Brazil…), which should be used to explain the crises or booms, not to compare.
6. Appendix A: Graphics should be incorporated into the text, can be made smaller.
Author Response
Reviewer 2
I thank the Publisher for sending me this work, which demonstrates extensive knowledge of the sector and of the data series, being a very interesting proposal. However, the work suffers from a number of important issues:
- The number of keywords is excessive, it must focus the interest of the article.
The keywords have been reduced from 9 to 6 focusing on keywords more in line with the topic.
- It has structural deficiencies:
- The abstract does not follow the indications of the instructions for authors (https://www.mdpi.com/journal/sustainability/instructions), its reading must clearly indicate the content of the work.
The abstract has been edited to include more direct results of the research.
- The introduction is disorganized, it is the result of the juxtaposition of elements, without the different paragraphs being related to each other. You have to follow a clear structure (build a speech) that goes from the general to the specific. For example (not necessarily, the author can search for another order that he considers to have scientific potential): a) Forests and crops in temperate areas; b) agricultural booms and crises (it is necessary to elaborate on their causes, without forgetting the thematic axis: Sustainable Agricultural Development Economics and Policy); c) North America/United States; d) Maine (this could be another heading or appear as a study area in the Methodology). It must have a structure according to results and discussion/conclusion.
The introduction has been re-organized and re-focused as requested on agricultural booms and busts and how this has coincided with crop diversification peaking in the middle of the 20th century and declining since in the U.S. Four paragraphs related to specialization and diversification pathways and impacts in the U.S. and in other countries were deleted along with associated citations.
- The results must coincide in the structure with what is stated in the methodology and the introduction (correlation of the parts).
- Deficiencies in the methodology: it is clear from the results that the author masters the methodology, but it is not properly explained. It should not start from what the reader knows about the subject, he should present it in the following way: "we take this data and apply these methods". On the other hand, the subheadings must coincide with what is done (analysis of Agricultural Booms and Busts…). I should comment on the sources he uses (and the methodologies used), which he only cites.
The methods section has been better introduced by adding 1) an introductory sentence in sub-section 2.2. Determining Historical Agricultural Production and Value and 2) an introductory paragraph for the sub-section 2.2. Calculating Agricultural Diversity Indicators. Also, the sub-section headings were simplified and clarified as requested focusing on the methodology of determining historical production and value of agricultural products and calculating diversity indicators.
- Content deficiencies:
- Although the introduction addresses the causes of the changes that occur, the discussion/conclusions are excessively descriptive (comparative) and do not go into the reasons why the changes occur (international competition, trade agreements, etc.). This must be solved by giving a new approach to the introduction (focused on analysis methodologies), or by introducing causality in the discussion/conclusions.
The re-focus of the Introduction outlined previously results in more focus on prior literature on analysis on diversification indicators in the U.S. and the New England region. By increasing the clarity of the Discussion section by removing the comparative part of this section, there is more emphasis and clarity on discussing the reasons why these changes occur so that causality is added to the discussion/conclusions.
- You should avoid decontextualized quotes, which do not coincide with the object of study or with the context. Talking about the boom and crisis of production (lines 46-53) in the European Neolithic to introduce a capitalist economic cycle is inappropriate, for several reasons: a) it is based on specific studies carried out with archaeological techniques (obviously different sources); b) the modern and contemporary production crises (the potato crisis in Ireland or the great famine...) are documented (even some of them, statistically) and coincide (not in causes, but in development) with the cycles of which the author speaks; c) there are archaeological studies on pre-colonial America, which may be more related to the work (but do not). On the other hand, there must be a temporal continuity, we cannot jump from the Neolithic to the date for which we have data.
The decontextualized parts of this paragraph in the Introduction and citations related to the Neolithic have been removed from the Introduction.
- Question of sources: in the discussion/conclusion sources are forced to make comparisons, which do not coincide neither in time nor in context with the results obtained. If we talk about the production crisis in Maine, we must contextualize it in a similar production crisis, not in data obtained from elsewhere in different contexts (India, Brazil…), which should be used to explain the crises or booms, not to compare.
The diversity indicator comparisons to other countries and citations have been removed from the Discussion and Conclusions section.
- Appendix A: Graphics should be incorporated into the text, can be made smaller.
All graphs in the Appendix have been integrated into the main body of writing by creating figures with two graphs a) and b).
Reviewer 3 Report
This paper is well structured and used various available statistics as far as possible that were efficiently and scientifically combined into several understandable graphical figures along the discussion.
Here is three points I would like to suggest.
1.
now reference no. 40 is just mistake due to unnecessary intention. Please recheck consistency of reference orders throughout the paper.
2.
I think, especially for international audiences, some maps should be developed and added to reinforce area image and discussion.
I think the following very good atlas could be used. You can use some maps from it with necessary modifications and clear source reference explanation?
Historical Atlas of Maine
byhttps://www.amazon.com/dp/0891011250?language=en_US
3.
overall manuscript is too long, so author can simplify and minimize texts especially in introduction and discussion parts?
Author Response
Reviewer 3
1.
now reference no. 40 is just mistake due to unnecessary intention. Please recheck consistency of reference orders throughout the paper.
Thank you or pointing this out and the unintended number of 40 in the left margin in the References section has been deleted.
- I think, especially for international audiences, some maps should be developed and added to reinforce area image and discussion.
I think the following very good atlas could be used. You can use some maps from it with necessary modifications and clear source reference explanation?
Historical Atlas of Maine
by Stephen J. Hornsby et al Eds
https://www.amazon.com/dp/0891011250?language=en_US
A figure map of Maine and the major agricultural production areas has been added as Figure 1a. According to a former Maine Department of Agriculture official, John Harker, this map is representative of the 1900’s / 1950’s to present as these regions of agricultural production have not changed since this time. The map was created by Maine Department of Agriculture (MDA) and MDA has given me permission to use his map. This type of map is better suited to show agriculture in the state of Maine, whereas the historical maps are focused more on townships and regions in Maine and are not focused on agriculture. We have also included as part of this Figure 1b as a GIS generated map using USDA Cropland Data Layer (CDL) where we mapped 9 crop categories from this publicly available CDL data set.
- overall manuscript is too long, so author can simplify and minimize texts especially in introduction and discussion parts?
The manuscript has been focused better and shortened, especially in the introduction and discussion sections.
Round 2
Reviewer 2 Report
This reviewer believes that the author's review demonstrates his effort to adequately respond to what was requested. The result is an article quite focused on the subject and interesting.
There is only one question, which does not affect the quality of the article. When the text is revised, it must be done with change control, in this case the deleted parts of the text do not appear. This forces the reviewer to read both texts (original and review) simultaneously, and complicates the review quite a bit. The author should take this into account for future articles.